# RoboOmni: Actions Are Just Another Modality for Vision-Language Models

**Dong Wang** [1 2]  **Zilong Chen** [1 2]  **Jirong Liu** [3]  **Ziqing Qiao** [1]  **Xin Xiao** [4]  **Bingyi Kang** [4]  **Hongtao Wu** [4]  **Xiao Ma** [4]  **Tao Kong** [4]  **Huaping Liu** [1 2]

## Abstract

Integrating Vision-Language Models (VLMs) into robotics has facilitated the development of generalizable Vision-Language Action (VLA) policies. However, unified discrete frameworks lag behind decoupled continuous designs due to limitations in action chunking and temporal modeling. To address this, we introduce `RoboOmni`, a unified multi-modal next-token prediction framework. Challenging the assumption that continuous modeling is essential for high-performance manipulation, `RoboOmni` demonstrates that *actions are just another modality* capable of being effectively modeled discretely. At the core of our method is Multi-Token Action Prediction (MTAP), which integrates action chunking directly into the discrete tokenizer. By preserving the native VLM training and inference pipeline, `RoboOmni` naturally benefits from large-scale multimodal co-training and modern decoding optimizations. Extensive evaluations on the CALVIN, SimplerEnv, and real-world platforms confirm that `RoboOmni` establishes new state-of-the-art performance, significantly outperforming diffusion-based baselines such as $\pi_0$. Notably, combining our proposed MTAP with the FAST tokenizer achieves a 94.4% average success rate on CALVIN, while the Bin tokenizer implementation attains a $27\times$ inference speedup compared to OpenVLA.

## 1. Introduction

The integration of powerful foundation models, especially Vision-Language Models (VLMs), into robotics is paving the way for Vision-Language-Action (VLA) systems capable of complex multimodal understanding and physical interaction (Zitkovich et al., 2023; Li et al., 2023; Shao & Xie, 2024; Lin et al., 2022; Tan et al., 2023). These models hold the promise of creating generalist robots that perform diverse manipulation tasks and generalize robustly across varied settings (Team et al., 2025). However, a critical challenge has emerged: while built upon highly capable VLMs, many current VLA implementations struggle to retain the broad generalization abilities inherent in their parent models. Instead, they often overfit significantly to the specific robotic datasets and environments seen during training (Li et al., 2026; Kim et al., 2024), losing the zero-shot or few-shot adaptability expected from foundation models and requiring costly retraining for new scenarios (Peng et al., 2023; Touvron et al., 2023).

The generalization gap between the VLM backbone and the downstream VLA is tied to the underlying architectural design and training paradigm (Li et al., 2026). Most VLAs apply VLMs as their feature extractors and feed representations into a *decoupled* continuous policy head, e.g., diffusion or flow policies (Team et al., 2024; Liu et al., 2024b), for action prediction. Although being effective for modeling continuous spaces, the decoupled approach separates action generation from core VLM reasoning and deviates from the pretrained internet-scale data.

In this paper, we argue that actions are just another modality for VLMs, and an unified next-token prediction framework captures the most underlying dependencies across all modalities, including actions. Prior approaches have explored this formulation (Kim et al., 2024; Pertsch et al., 2025), but their performance struggles compared with the decoupled approaches. The root cause lies with the fundamental auto-regressive training paradigm: the single-step action token generation causes severe compounding error during inference in a Markov Decision Process (MDP), and it further slows the inference speed compared with decoupled approaches, where an action chunk consisting of multi-step actions is being generated in a single forward pass. As a result, unified approaches often run with a single-step history and fail to fully utilize the rich information of past observations and actions.

To overcome these limitations, we present `RoboOmni`, a

---

[1]Department of Computer Science and Technology, TNLIST, Tsinghua University, Beijing, China [2]Institute for Embodied Intelligence and Robotics, Tsinghua University, Beijing, China [3]Shanghai Jiao Tong University, Shanghai, China [4]ByteDance, Beijing, China. Correspondence to: Huaping Liu <hpliu@tsinghua.edu.cn>.

*Proceedings of the $43^{rd}$ International Conference on Machine Learning*, Seoul, South Korea. PMLR 306, 2026. Copyright 2026 by the author(s).

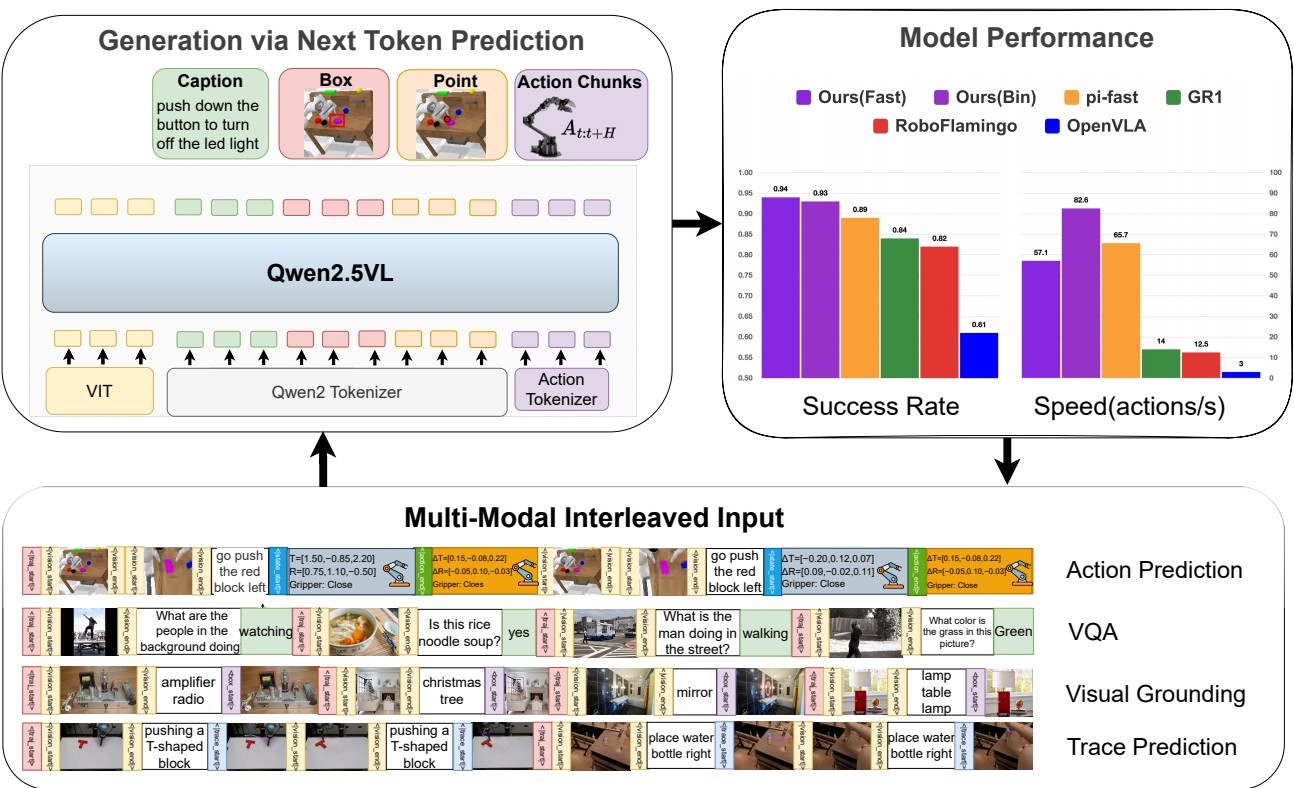

*Figure 1.* Overview of the `RoboOmni` framework and its performance. The bottom section illustrates the multi-modal interleaved data input. The top-left section details the model architecture, which processes multi-modal interleaved inputs to produce various outputs. The right section displays SOTA performance of `RoboOmni` on the CALVIN benchmark and its inference speed. `RoboOmni` is approximately 27x faster than the unified approach OpenVLA and 6.6x faster than the decoupled approach RoboFlamingo.

generalist robotic policy model that interleaves vision, language, and action tokens within a unified next-token prediction framework. This design enables long-context, multi-modal co-training and allows the model to explicitly reason over historical observations and actions. Our experiments show that such interleaved, long-context multi-modal training significantly improves performance and generalization, highlighting the importance of both temporal context and cross-modal fusion. Our key technical contribution is a discrete temporal modeling strategy that preserves VLM structures while enabling efficient multi-step action prediction. Specifically, we introduce **Multi-Token Action Prediction (MTAP)**, which performs parallel decoding of $H$ actions by repeating only the last layer for action tokens, inspired by (Gloeckle et al., 2024; Liu et al., 2024a). MTAP strikes a balance between minimal architectural modification and high action prediction accuracy, effectively enabling action chunking within the unified framework. Importantly, although incorporating long histories typically increases inference cost, `RoboOmni` does not suffer from this drawback. Thanks to its consistent VLM-style architecture, `RoboOmni` directly benefits from advanced optimization techniques developed for VLM serving, such as KV-caching and RadixAttention (Zheng et al., 2024a). As shown in

Figure 1, `RoboOmni` achieves an inference speed of 82.6 Hz with an action chunk size of 10 and a history length of 5, outperforming OpenVLA (Kim et al., 2024) by 27×, demonstrating that long-context multi-modal inference can be both expressive and efficient.

## 2. Related Work

### 2.1. Vision-Language-Action Models

Existing VLA models, designed for multimodal understanding and robotic interaction, can be categorized along several axes related to data processing. Key distinctions include whether models utilize temporal history (Li et al., 2023) or operate on single frames (Intelligence et al., 2025), how actions are represented (discrete tokens (Zitkovich et al., 2023; Kim et al., 2024) vs. continuous vectors (Liu et al., 2024b; Team et al., 2024)), and whether they predict actions step-by-step or employ action chunking (Zhao et al., 2023).

Architecturally, diverse training paradigms are employed. Some models are built on diffusion policies, either trained specifically for robotics like Octo (Team et al., 2024) or integrated within larger systems like RDT-1B (Liu et al., 2024b). A dominant approach adapts powerful pre-trained VLMs

as backbones, finetuning them with robotics data, as seen in RT-2 (Zitkovich et al., 2023), RoboFlamingo (Li et al., 2023), and OpenVLA (Kim et al., 2024). Hybrid strategies also exist, such as $\pi_0$ (Black et al., 2024), which combines VLM encoding with diffusion-based action decoding.

Analyzes like RoboVLMs (Li et al., 2026) often suggest that continuous action representations, processed with historical context via separate decoder heads, yield optimal results, reinforcing the view that action generation is primarily a regression task ill-suited for the next-token prediction common in language modeling. However, we demonstrate that RoboOmni, by integrating the advanced optimization techniques from VLMs, not only unifies the modalities, but provides stronger performances than decoupled models, comparably fast inference speed, and better scalability.

## 2.2. Action Co-training with Vision-Language Tasks

Beyond optimizing the core action generation process, enhancing VLA capabilities through co-training with auxiliary vision-language (VL) tasks has become a significant research thrust. This strategy aims to imbue VLAs with richer semantic understanding, improved reasoning, and better generalization by exposing them to related, non-robotic objectives during training. Early evidence highlighted the benefits of general VL dataset co-training alongside discrete action prediction, with RT-2 demonstrating improved adaptation to novel objects through this approach (Zitkovich et al., 2023). Subsequent studies have investigated incorporating intermediate representations that bridge vision and action more explicitly. VLAs such as LLaRVA (Zhang et al., 2023), Hamster (Li et al., 2025), and TraceVLA (Zheng et al., 2024b) utilize future visual trace prediction as an auxiliary objective to foster better vision-action alignment. An alternative direction involves learning latent action representations from large-scale human video datasets, as pursued by LAPA (Ye et al., 2024), aiming to mitigate the domain gap between human demonstrations and robot execution. Further efforts targeting higher-level cognitive skills have seen models like $\pi_{0.5}$ (Intelligence et al., 2025) and Gemini Robotics (Team et al., 2025) integrating specific auxiliary objectives related to high-level task planning and object detection, explicitly enhancing planning capabilities and spatial understanding.

## 3. RoboOmni

RoboOmni fundamentally reconceptualizes the integration of action capabilities into VLMs. Our approach is driven by the objective to minimally alter established VLM architectures while seamlessly incorporating the action modality. We achieve this by structuring the input as multi-modal interleaved sequences of vision, language, state, and action tokens and using Multi-Token Action Prediction (MTAP)

for action chunking. This allows the prediction of multiple future action steps without modifying the inherent causal attention mechanisms.

We formalize the manipulation task as a sequence modeling problem. The policy $\pi$ learns to generate a chunk of $H$ future actions, $a_{t:t+H-1}$, to complete a task specified by a language instruction $l \in \mathcal{L}$. The policy's decision is conditioned on a history $h_t$ that includes recent visual observations $o_t \in \mathcal{O}$, proprioceptive states $s_t \in \mathcal{S}$ (*e.g.*, end-effector pose), and past actions from the action space $\mathcal{A}$. By tokenizing all modalities into a unified sequence, we train the model using a standard causal, next-token prediction objective. This directly aligns with how contemporary VLMs are trained, enabling RoboOmni to benefit from VLM optimization techniques (*e.g.*, inference optimizations, multimodal pre-training) to significantly improve generalization and efficiency.

### 3.1. MTAP for Action Chunking

Action chunking, or predicting multiple future actions simultaneously, is a key technique to improve the performance and sample efficiency of robot policies (Zhao et al., 2023; Pertsch et al., 2025; Li et al., 2026). When integrating this capability into a VLM-based causal transformer backbone, it typically requires generating action sequences whose length is proportional to the chunking window. However, such extended sequential prediction in causal architectures often suffers from compounding errors and increased inference latency, hindering true long-horizon anticipation. To overcome these challenges, we introduce a versatile Multi-Token Action Prediction (MTAP) framework that enables efficient, parallelized action prediction within a unified discrete architecture. Currently, action discretization follows two primary paradigms: (1) **Binning-based** methods, which discretize each action dimension independently (Kim et al., 2024), and (2) **Frequency-space** methods, such as FAST (Pertsch et al., 2025), which represent entire action trajectories in the spectral domain. By abstracting the prediction objective, MTAP serves as a unified solution capable of accommodating both of these distinct tokenizer archetypes.

**Binning-based Action Tokenization.** For tokenizers that discretize each action step independently (Binning), the sequence retains complete physical information but suffers from high temporal redundancy. We adapt the method from (Gloeckle et al., 2024) to perform **temporal compression**. Specifically, let $h_{last}$ denote the hidden state from the shared transformer backbone corresponding to the last input token at timestep $t$. To predict an action chunk of size $H$, we employ $H$ parallel projection heads (implemented as lightweight MLPs). Each head $k \in \{0, \ldots, H-1\}$ maps the shared representation $h_{last}$ to a distinct latent state $z_k$. Each state $z_k$ is then passed through a *shared* language model head (LMHead) to produce logits for the future action

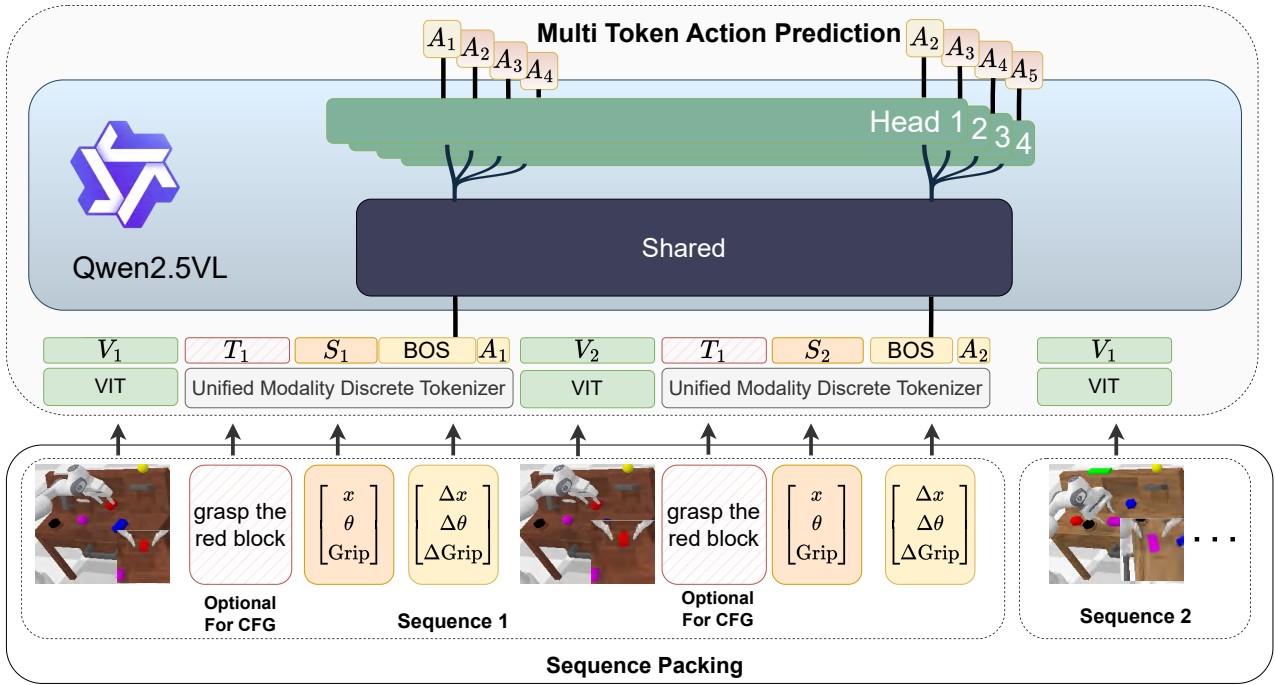

*Figure 2.* Architectural overview of `RoboOmni`. The model processes multi-modal interleaved input sequences comprising visual observations ($V$), text instructions ($T$), robot states ($S$), and actions ($A$). These sequences are packed for efficiency, where the text instructions ($T$) are optionally masked as part of Classifier-Free Guidance (CFG) training. `RoboOmni` supports MTAP through shared layers, followed by parallel heads, enabling the prediction of action chunks.

at relative step $k$, *i.e.,* $a_{t+k}$. This design enables parallel decoding of the entire action chunk from a single shared context, aggregating the loss across predictions:

$$\mathcal{L} = \sum_{k=0}^{H-1} \mathcal{L}_{CE}(\text{LMHead}(z_k), a^*_{t+k}) \qquad (1)$$

where $a^*_{t+k}$ represents the ground-truth token for action $a_{t+k}$. This parallel structure effectively mitigates the compounding error inherent in sequential decoding and drastically accelerates inference.

**Frequency-space Action Sequence Tokenization (FAST).** For advanced tokenization schemes like FAST (Pertsch et al., 2025), the action chunk is already transformed into a variable-length, frequency-domain token sequence. This transformation breaks the explicit step-by-step temporal correspondence. Consequently, we adapt MTAP to align with standard multi-token prediction (Liu et al., 2024a) as a mechanism to enhance backbone convergence. Here, the objective is to predict the next $H$ *tokens* in the frequency sequence. For the hidden state $z_j$ of an input token $y_j$, we employ $H$ parallel prediction layers. The $k$-th pathway is trained to predict the token at position $j + k + 1$. The loss

function reflects this token-index-based objective:

$$\mathcal{L} = \sum_{j} \sum_{k=0}^{H-1} \mathcal{L}_{CE}(\text{LMHead}(z_{j,k}), y^*_{j+k+1}) \qquad (2)$$

where $z_{j,k}$ is the $k$-th hidden state for token $y_j$ and $y^*_{j+k+1}$ is the ground-truth future token.

**Strategic Implementation of MTAP.** A critical insight of `RoboOmni` is that the role of MTAP shifts based on the tokenization scheme. For **Bin tokenization**, the raw action sequence is lossless but long. MTAP here acts as a **temporal compressor**: by utilizing parallel heads to predict the chunk simultaneously, we trade a marginal increase in parameter count (parallel heads) for a massive gain in inference speed (reducing $H$ forward passes to 1) while maintaining lossless precision. In contrast, **FAST tokenization** inherently performs lossy compression via the Fourier transform. Therefore, we employ MTAP primarily as an auxiliary **training objective** to facilitate backbone modeling of the complex frequency tokens, rather than solely for inference acceleration.

### 3.2. Multi-Modal Action Co-Training

To support multi-modal co-training, we build a unified tokenization scheme that encodes all modalities. These in-

clude **Visual** inputs, **Text** inputs, **Bounding Box** and **Pixel Point**, as well as **Robot State** and **Action** modalities. All are mapped into a shared representational space (see Appendix A for tokenization details of each modality). We incorporate several VL co-training tasks to enhance the capabilities of the model:

**Visual Grounding.** We include a grounding objective to strengthen spatial understanding and object localization by predicting text-tokenized bounding box coordinates for referred objects.

**Point Trace Prediction.** To encourage short-horizon temporal reasoning and motion understanding, we introduce a 2D end-effector trace prediction task inspired by (Li et al., 2025). The model predicts tokenized pixel trajectories obtained by projecting 3D gripper coordinates onto the image plane, using partial-trajectory conditioning for robustness.

**Visual Question Answering (VQA).** We incorporate VQA-style objectives to preserve and enhance general image understanding, multimodal reasoning, and instruction-following.

Dataset details and collection procedures are provided in Appendix B.

### 3.3. Training VLA as VLM

One of the core advantages of `RoboOmni` is its unified representation of action and all other modalities, which allows for the seamless integration of VLM optimization techniques with VLA training. We employ several advanced training strategies designed to enhance the stability, efficiency, and predictive capabilities of the next-token prediction framework for robotics.

**Optimize `RoboOmni` as VLMs (including sequence packing).** A significant limitation of VLA policies, particularly those using discrete action tokenization (Kim et al., 2024; Zitkovich et al., 2023), is their reliance solely on the current observation $o_t$, thereby ignoring history. `RoboOmni` preserves a standard VLM-style next-token prediction backbone, where each trajectory is represented as an interleaved sequence of past observations ($o_{t-T:t}$), robot states ($s_{t-T:t}$), and actions ($a_{t-T:t-1}$) up to the current timestep $t$.

To improve training efficiency under highly variable sequence lengths, we adopt sequence packing (Krell et al., 2021) as a VLM training technique. Importantly, while tokens within each trajectory remain interleaved, we allow visibility across packed trajectories by omitting attention masks between packed samples. This encourages the model to infer task and dynamics from immediate context rather than memorizing single trajectories, which empirically improves convergence. During inference, we utilize modern LLM serving platforms, such as SGLang (Zheng et al., 2024a), to accelerate inference.

**Classifier-Free Guidance Training.** To improve robustness and leverage heterogeneous data, we adopt Classifier-Free Guidance (CFG) (Ho & Salimans, 2022). During training, we randomly drop language instruction tokens $l$ with probability $0.2$. This encourages the model to predict actions from visuomotor context alone while also enabling learning from demonstrations without language annotations, broadening the data distribution and improving long-horizon policy reliability.

By jointly optimizing for these diverse objectives alongside the primary action prediction task, the model learns more robust and generalizable representations. (See details in Appendix B)

## 4. Experiment

We evaluate `RoboOmni` across three complementary settings: (1) long-horizon multi-task manipulation on the CALVIN benchmark, (2) Google Robot tasks in the SimplerEnv simulator, and (3) real-world robot experiments.

Across all settings, we compare `RoboOmni` against three representative and widely adopted unified VLA baselines: **OpenVLA** (Kim et al., 2024) is an autoregressive vision-language-action (VLA) model that predicts discretized action tokens from a single frame, without historical context or vision-language co-training. In our reproduction, the model is trained exclusively on manipulation data to align with the setting without VLM described in the original paper. **RoboVLM** (Li et al., 2026) represents a decoupled design where a VLM backbone feeds into a dedicated policy head for continuous action decoding. $\pi_0$**-FAST** (Pertsch et al., 2025) combines a VLM with the FAST tokenizer. Our reproduction uses a PaliGemma-3B backbone and is trained with the identical data mixture as `RoboOmni`. See Appendix D.2 for implementation details.

### 4.1. Evaluation on Calvin

CALVIN (Mees et al., 2022b) is a simulation benchmark for multi-task tabletop manipulation. It comprises four scene splits (A, B, C, and D) covering 34 distinct manipulation tasks and contains 22,966 human-teleoperated demonstrations annotated with natural language instructions. Following prior work, we train on the ABCD and ABC splits and evaluate solely on split D with 1,000 rollouts per model. We report the success rates of achieving 1 through 5 consecutive tasks, as well as the average number of tasks completed per trial (Task Len.).

**Unified discrete-token SOTA.** As shown in Table 1, `RoboOmni` establishes a new state-of-the-art, challenging the prevailing notion that continuous actions are required for high-performance manipulation. Specifically,

*Table 1.* Performance comparison on the **CALVIN** benchmark. The table evaluates models on two settings: in-distribution performance (Train: ABCD, Eval: D) and out-of-distribution generalization (Train: ABC, Eval: D). Our `RoboOmni` models, with both Bin and FAST tokenizers, establish new state-of-the-art (SOTA) results in both settings. **Bold** indicates the best performance, and underline indicates the second-best among all methods.

| Method | Train | Consecutive tasks success rates | | | | | Task Len. |
|---|---|---|---|---|---|---|---|
| | | 1 | 2 | 3 | 4 | 5 | |
| MCIL (Lynch & Sermanet, 2020) | ABCD | 0.373 | 0.027 | 0.002 | 0.000 | 0.000 | 0.40 |
| R3M (Frozen) (Nair et al., 2022) | | 0.085 | 0.005 | 0.001 | 0.000 | 0.000 | 0.10 |
| Voltron (Frozen) | | 0.101 | 0.003 | 0.001 | 0.000 | 0.000 | 0.11 |
| Voltron (Fine-tuned) | | 0.837 | 0.566 | 0.352 | 0.208 | 0.115 | 2.08 |
| RT-1 (Brohan et al., 2022) | | 0.844 | 0.617 | 0.438 | 0.323 | 0.227 | 2.45 |
| OpenVLA (Kim et al., 2024) | | 0.921 | 0.732 | 0.565 | 0.455 | 0.346 | 3.03 |
| HULC (Mees et al., 2022a) | | 0.889 | 0.733 | 0.587 | 0.475 | 0.383 | 3.06 |
| RoboFlamingo (Li et al., 2023) | | 0.964 | 0.896 | 0.824 | 0.740 | 0.662 | 4.09 |
| GR-1 (Wu et al., 2023) | | 0.949 | 0.896 | 0.844 | 0.789 | 0.731 | 4.21 |
| UP-VLA (Zhang et al., 2025) | | 0.962 | 0.921 | 0.879 | 0.842 | 0.812 | 4.42 |
| RoboVlMs (Li et al., 2026) | | 0.967 | 0.930 | 0.899 | 0.865 | 0.826 | 4.49 |
| $\pi_0$-FAST (Pertsch et al., 2025) | | 0.974 | 0.936 | 0.892 | 0.848 | 0.803 | 4.45 |
| `RoboOmni`(Bin) | | 0.997 | 0.973 | 0.940 | 0.895 | 0.834 | 4.64 |
| `RoboOmni`(FAST) | | **0.997** | **0.982** | **0.951** | **0.918** | **0.881** | **4.73** |
| Voltron (Frozen) | ABC | 0.026 | 0.001 | 0.000 | 0.000 | 0.000 | 0.03 |
| Voltron (Fine-tuned) | | 0.569 | 0.272 | 0.105 | 0.038 | 0.014 | 1.00 |
| RT-1 | | 0.533 | 0.222 | 0.094 | 0.038 | 0.013 | 0.90 |
| HULC | | 0.418 | 0.165 | 0.057 | 0.019 | 0.011 | 0.67 |
| GR-1 | | 0.854 | 0.712 | 0.596 | 0.497 | 0.401 | 3.06 |
| UP-VLA | | 0.928 | 0.865 | 0.815 | 0.769 | 0.699 | 4.08 |
| $\pi_0$-FAST (Pertsch et al., 2025) | | 0.989 | 0.929 | 0.842 | 0.777 | 0.698 | 4.24 |
| RoboVLMs | | 0.980 | 0.936 | 0.854 | 0.778 | 0.704 | 4.25 |
| `RoboOmni`(Bin) | | 0.988 | 0.933 | 0.860 | 0.795 | 0.721 | 4.30 |
| `RoboOmni`(FAST) | | **0.992** | **0.941** | **0.882** | **0.804** | **0.735** | **4.35** |

`RoboOmni`(FAST) achieves an **88.1%** 5-task success rate, significantly outperforming decoupled continuous baselines like RoboVLMs (82.6%) and the concurrent unified model $\pi_0$-FAST (80.3%). This validates that a properly optimized discrete-token framework can surpass continuous counterparts.

**Architectural Effectiveness and Generalization.** Our controlled comparisons isolate the benefits of the `RoboOmni` architecture. `RoboOmni`(Bin) improves over the single-frame OpenVLA by nearly **50%** (83.4% vs 34.6%), confirming the criticality of historical context and our MTAP action chunking. Furthermore, `RoboOmni`(FAST) outperforms $\pi_0$-FAST by roughly 8% using identical data and tokenizers, proving that our performance gains stem from architectural design rather than tokenization alone. Notably, the FAST variant exhibits superior out-of-distribution generalization (ABC→D), suggesting the frequency-domain representation effectively offloads temporal modeling pressure from the backbone.

### 4.2. Evaluation on SimplerEnv

We evaluate `RoboOmni` on the Google Robot tasks within **SimplerEnv** (Li et al., 2024), which is designed to assess real-to-sim transfer for VLAs trained on real-world data. In our setting, models are trained on a subset of the Open X-Embodiment (OXE) (Collaboration et al., 2025) dataset and evaluated in the *Visual Matching* protocol. SimplerEnv provides realistic simulated counterparts for a subset of OXE scenarios, enabling controlled evaluation of policies trained from real demonstrations under visually matched simulation. See Appendix D.3 for detailed implementation settings.

**State-of-the-art Real-to-Sim Transfer.** As presented in Table 2, `RoboOmni` establishes a new performance standard, significantly outperforming prior unified and continuous baselines. `RoboOmni`(FAST) achieves an average success rate of **86.8%**, surpassing the strongest continuous baseline, SpatialVLA, by over **16%**. Notably, our discrete-token approach also outperforms $\pi_0$-FAST (61.9%), further confirming that our unified architecture effectively bridges the gap

*Table 2.* **Real-to-Sim performance comparison on Google Robot tasks in SimplerEnv (Visual Matching setting).** We report the average success rate over 3 distinct tasks. `Pick Can` aggregates results across horizontal, vertical, and standing orientations; `Drawer Interaction` includes both open and close drawer tasks. `RoboOmni` demonstrates superior robustness to visual domain shifts compared to baselines.

| Method | Pick Can | Move Near | Drawer Interaction | Avg |
|---|---|---|---|---|
| Octo-Base | 0.170 | 0.042 | 0.227 | 0.146 |
| OpenVLA | 0.163 | 0.462 | 0.356 | 0.327 |
| RT-2-X | 0.787 | 0.779 | 0.250 | 0.605 |
| RoboVLMs | 0.727 | 0.663 | 0.268 | 0.553 |
| $\pi_0$-FAST | 0.753 | 0.675 | 0.429 | 0.619 |
| SpatialVLA | 0.810 | 0.696 | 0.593 | 0.700 |
| `RoboOmni`(Bin) | 0.957 | 0.849 | 0.694 | 0.833 |
| `RoboOmni`(Fast) | **0.970** | **0.917** | **0.719** | **0.868** |

between discrete VLM generation and continuous robotic control.

**Precision and Visual Robustness.** These results highlight two critical strengths of `RoboOmni`. First, the high success rates on object-interaction tasks (e.g., *Pick Coke Can*) demonstrate that our discrete tokenization, coupled with MTAP action chunking, achieves the fine-grained precision required for short-horizon manipulation—traditionally a stronghold of continuous policies. Second, the model's dominance in the *Visual Matching* setting indicates exceptional robustness to sim-to-real visual shifts, suggesting that preserving the VLM's pre-trained visual representations enables superior generalization across visual domains compared to decoupled or scratch-trained encoders.

### 4.3. Real Robot Experiments

To validate the effectiveness of `RoboOmni` in physical environments, we conducted extensive experiments on a real robot platform featuring a Kinova Gen-3 arm. The training dataset consists of 18k human demonstrations across 37 tasks, including both pick-and-place and non-pick-and-place manipulation. We evaluate performance across four settings designed to test generalization: **Simple** (seen scenarios), **Unseen Distractors**, **Unseen Instructions** (synonyms generated by GPT-4), and **Unseen Objects** (manipulating novel objects). We compare `RoboOmni` against strong baselines including OpenVLA (Kim et al., 2024), Octo (Team et al., 2024), GR-1 (Wu et al., 2023), RoboVLMs (Li et al., 2026), and $\pi_0$-FAST (Pertsch et al., 2025).

**Robust Generalization to Novel Scenarios.** As illustrated in Figure 3, `RoboOmni` demonstrates superior performance across all evaluation metrics. In the **Simple** setting, our model achieves a **93%** success rate, establishing a strong

*Table 3.* Ablation study of MTAP and different tokenizers.

| Settings | | Top K Success Rate | | Task Len. | Speed (ms/action) |
|---|---|---|---|---|---|
| MTAP | Tokenizer | Top 1 | Top 5 | | |
| ✓ | FAST | 0.997 | 0.881 | 4.73 | 17.5 |
| ✓ | BIN | 0.997 | 0.834 | 4.64 | 12.1 |
| ✗ | FAST | 0.990 | 0.801 | 4.52 | 24.2 |
| ✗ | BIN | 0.990 | 0.679 | 4.24 | 107 |

baseline for precise control. Crucially, `RoboOmni` exhibits exceptional robustness in the most challenging **Unseen Objects** setting, maintaining a **91%** success rate. In contrast, the second-best baseline, $\pi_0$-FAST, drops to 61%, and other models suffer more severe degradation. This indicates that our unified architecture and co-training strategy enable a deep, transferable understanding of manipulation skills rather than overfitting to training instances. On average, `RoboOmni` achieves a **91%** success rate, significantly surpassing $\pi_0$-FAST (68%) and RoboVLMs (60%). For a comprehensive breakdown of the experimental setup, task list, and detailed quantitative results, please refer to Appendix E.

### 4.4. Ablation Study

We conduct a series of ablation studies to evaluate the contributions of key components in our framework on Calvin Benchmark. Unless otherwise specified, all experimental settings follow those detailed in Section 4.1.

*Table 4.* Ablation study on the number of bins for action discretization. All models are trained with MTAP. The default setting used in our main experiments is 256 bins.

| Tokenizer | Bin Size | Top 1 | Top 3 | Top 5 | Task Len. |
|---|---|---|---|---|---|
| FAST | 128 | 0.996 | 0.950 | 0.861 | 4.70 |
| | 256 | **0.997** | **0.951** | **0.881** | **4.73** |
| | 1024 | 0.990 | 0.940 | 0.871 | 4.68 |
| BIN | 128 | 0.989 | 0.920 | 0.837 | 4.59 |
| | 256 | 0.997 | 0.940 | 0.834 | 4.64 |
| | 1024 | 0.980 | 0.888 | 0.790 | 4.44 |

**Impact of MTAP and Tokenizer.** Our primary ablation, presented in Table 3, investigates the impact of Multi-Token Action Prediction (MTAP). The results clearly demonstrate that MTAP is broadly effective, providing a substantial performance boost for both tokenizer schemes. For the FAST tokenizer, enabling MTAP improves the 5-task success rate from 80.1% to **88.1%**. The effect is even more pronounced for the Bin tokenizer, where MTAP elevates the success rate dramatically from 67.9% to **83.4%**. Interestingly, MTAP also reverses the inference speed characteristics of the tokenizers. Without MTAP, the fully autoregressive Bin tokenizer is exceedingly slow (107 ms/action). By enabling parallel decoding over the action chunk, MTAP provides a near-linear speedup, slashing the inference time to just **12.1**

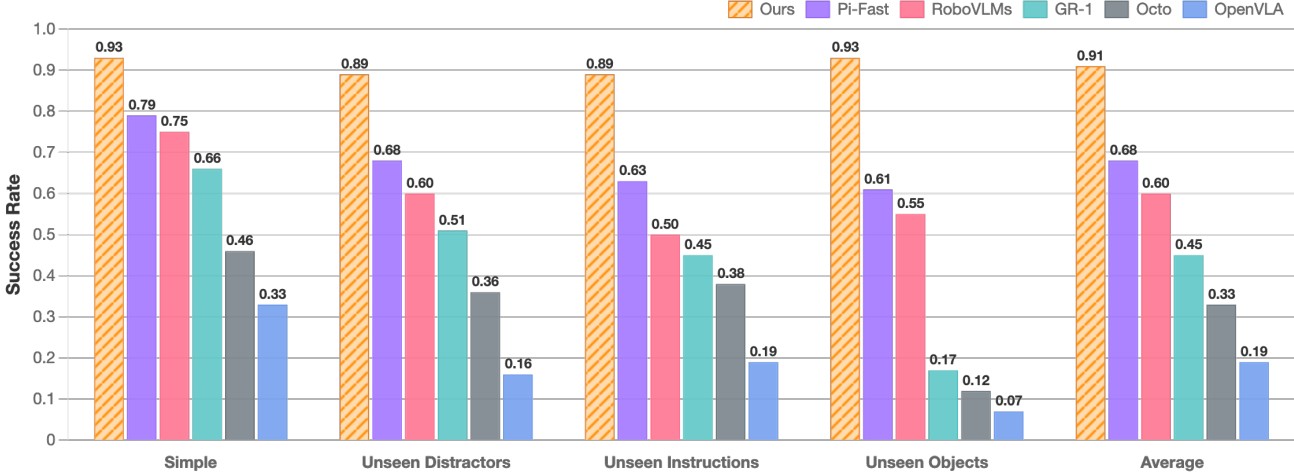

*Figure 3.* Comparison of success rates in the real-world setting. `RoboOmni` consistently outperforms baselines, including $\pi_0$-FAST and RoboVLMs, particularly in the challenging Unseen Objects setting.

**ms/action**. This not only makes the Bin tokenizer significantly more efficient but also faster than the MTAP-enabled FAST tokenizer (17.5 ms/action), presenting a compelling trade-off between peak performance and inference speed. We conducted extensive ablation studies on the CALVIN benchmark to systematically evaluate the contribution of each key design choice within the `RoboOmni` framework.

**Impact of Action Bin Size.** We investigate how the precision of action discretization affects policy performance. As detailed in Table 4, we evaluate bin sizes of 128, 256, and 1024 for both FAST and Bin tokenizers. For the FAST tokenizer, performance peaks with 256 bins, achieving a **88.1%** 5-task success rate. Using a coarser discretization of 128 bins leads to a slight decline (86.1%), suggesting a potential loss of necessary precision for fine-grained movements. Conversely, a finer granularity of 1024 bins also results in a performance drop (87.1%), likely because it increases the complexity of the prediction task without a proportional benefit in control accuracy. A similar trend is observed for the Bin tokenizer, where performance is highest with 128 bins (83.7%) and 256 bins (83.4%), but degrades significantly when using 1024 bins (79.0%). This suggests that for both methods, 256 bins provides the best trade-off between expressive action representation and learnability.

**Ablation on Architectural and Training Components.** Based on its compelling trade-off between performance and efficiency, we use the `RoboOmni`(Bin) configuration as the default for our final ablation studies, presented in Table 5. Our analysis shows that each component is crucial for final performance. Ablating the **history length** reveals that increasing the window size from 1 to 5 yields a significant performance gain (81.3% to 83.4% 5-task success rate), while a further increase to 10 offers diminishing returns. We also observe a clear scaling trend with **model size**,

*Table 5.* Ablation studies on window size, model size, and training strategies. The default setting is `RoboOmni`(Bin) with a window size of 5 and a 7B parameter model, trained with all components.

| Setting | Top 1 | Top 3 | Top 5 | Task Len. |
|---|---|---|---|---|
| *Default Configuration* | | | | |
| `RoboOmni`(Bin) | **0.997** | **0.940** | **0.834** | **4.64** |
| *Ablation on Window Size* | | | | |
| Window Size = 1 | 0.973 | 0.897 | 0.813 | 4.49 |
| Window Size = 10 | 0.985 | 0.914 | 0.824 | 4.55 |
| *Ablation on Model Size* | | | | |
| Qwen2-VL-2B | 0.981 | 0.886 | 0.776 | 4.42 |
| Qwen2.5-VL-3B | 0.984 | 0.911 | 0.819 | 4.54 |
| Qwen2-VL-7B | 0.982 | 0.918 | 0.828 | 4.57 |
| *Ablation on Training Strategies* | | | | |
| Without VLM Dataset | 0.991 | 0.911 | 0.806 | 4.53 |
| Without Sequence Packing | 0.983 | 0.897 | 0.791 | 4.46 |
| Without CFG | 0.987 | 0.897 | 0.795 | 4.48 |

where performance improves from 77.6% (2B) to 83.4% (7B). Finally, removing any of our core **training strategies** degrades performance. The exclusion of VLM data co-training, sequence packing, or Classifier-Free Guidance (CFG) all leads to a noticeable drop in task success, confirming that each strategy synergistically contributes to the robustness and capability of our final model.

## 5. Conclusion

We introduced `RoboOmni`, a unified multi-modal framework that treats actions as just another modality for VLMs. Our novel MTAP strategy enhances historical context integration and mitigates action distribution shift. By preserving the core VLM architecture, `RoboOmni` seamlessly incorporates advanced optimizations and multi-modal co-

training paradigms. Extensive evaluations on CALVIN and a real-world robot demonstrate state-of-the-art performance, proving a well-designed unified framework can outperform decoupled approaches. Future work will involve scaling our approach with larger VLM backbones, expanding co-training tasks to enhance physical reasoning, and investigating more sample-efficient adaptation to novel robotic platforms in diverse real-world settings.

## Acknowledgements

This work was jointly supported by the National Key Research and Development Program of China under grant 2024YFB4708000, National Natural Science Fund under grant 62120106005, Beijing Natural Science Foundation under grant no. L253006, and Fundamental and Interdisciplinary Disciplines Breakthrough Plan of the Ministry of Education of China under grant no. JYB2025XDXM109.

## Impact Statement

This paper advances unified VLA modeling for robotic manipulation by treating actions as another modality in a VLM-style next-token prediction framework. By enabling a single model to condition on interleaved vision, language, robot state, and action tokens, our approach can reduce system complexity and engineering overhead, potentially lowering the barrier to deploying general-purpose robot assistants in settings such as warehouse automation, home assistance, and lab operations.

At the same time, learning from large-scale real-world demonstrations (including subsets of Open X-Embodiment used in our experiments) may encode dataset biases, unsafe edge-case behaviors, and unintended correlations. Moreover, strong benchmark performance (e.g., CALVIN, SimplerEnv) does not guarantee safe operation in unstructured physical environments; failures can cause property damage or injury, and misuse is possible if manipulation capabilities are repurposed for harmful tasks.

To mitigate these risks, we recommend deploying such models only with hardware-level safety interlocks, conservative action/state constraints, and human supervision; performing platform-specific safety validation before any real-world use; and documenting training data provenance, known limitations, and intended use cases. Our results are reported on research benchmarks and controlled robot setups, and we do not claim readiness for unconstrained, safety-critical deployment.

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

## A. Unified Modality Representation

RoboOmni processes a diverse set of input modalities and is capable of generating various outputs, all within a unified next-token prediction framework. Inputs include visual information such as images and video streams, natural language instructions or captions, and proprioceptive robot states. The model can then generate sequences of actions for robotic manipulation, textual responses for tasks like Visual Question Answering (VQA), bounding boxes for visual grounding, and 2D point traces. To achieve this, all these relevant modalities—including language, vision, robot states, actions, points, and bounding boxes—are mapped into a shared discrete token sequence, as illustrated in Figure 4. This unification relies on the modality-specific tokenization schemes detailed below.

**Text Tokenizer:** Natural language instructions or captions are processed using the standard text tokenizer provided with the Qwen2.5-VL model. This tokenizer is utilized for both encoding textual inputs and decoding textual outputs generated by RoboOmni.

**Visual Representation:** Input images are processed by the Vision Transformer (ViT) component of the Qwen2.5-VL model. A key feature of this ViT is its support for variable resolution inputs, enabling it to seamlessly handle both static images and dynamic video streams. Visual features undergo temporal compression by a factor of 2. For spatial feature resolution reduction, a patch size of $14 \times 14$ is employed, combined with a subsequent pooling factor of 2. This results in an overall spatial compression factor of $28 \times 2$ relative to the input image dimensions. The output of this process is a sequence of visual tokens, which are encapsulated by special marker tokens: `<vision_start>` at the beginning and `<vision_end>` at the end of the visual token sequence.

**State and Action Tokenizer:** Continuous robot states and actions (typically represented as delta states), both comprising 6 Degrees of Freedom (DoF) end-effector poses (XYZ, RPY) and a gripper state, are discretized into token sequences. Following the methodology of OpenVLA, each continuous dimension is independently mapped to one of 256 discrete bins. Normalization is performed using the 1st and 99th percentiles of the distribution of that dimension observed in the training dataset; this range is assigned to $[-1, 1]$ before the binning process. This robust normalization approach avoids undue sensitivity to outliers that can affect standard min-max normalization.

A crucial distinction in our approach is the handling of the new action and state tokens. Instead of replacing low-frequency words in the existing vocabulary of the Qwen2.5-VL tokenizer, we extend its vocabulary. Specifically, we add 256 unique state tokens (named 'scode_0' through 'scode_255', as exemplified for a sequence in Figure 4)

and 256 unique action tokens (similarly, 'acode_0' through 'acode_255') to the vocabulary of the tokenizer. Each of these tokens corresponds to one of the discrete bin values. Critically, these new tokens are incorporated as *normal tokens* rather than *special tokens*. This design choice was informed by observations that the Qwen2.5-VL architecture applies specific internal processing to special tokens, which could introduce unintended complexities or instability during the training of the VLA model. The resulting sequences of these 'scode_X' (for state) or 'acode_X' (for action) tokens, one for each dimension of the state or action vector, are then prefixed by their respective special start tokens, namely `<|state_start|>` and `<|action_start|>`.

The selection of 256 bins per dimension offers a fine-grained discretization that provides sufficient precision for typical robotic control tasks. For instance, considering a representative action scale where a 2 cm end-effector movement is functionally significant, the discretization error per dimension would be less than $2\,\text{cm}/256 \approx 0.078$ mm. This level of error is considerably smaller than the inherent error margins of most low-level robot controllers and, as such, can be considered negligible for practical manipulation purposes.

**Point and Bounding Box Tokenizer:** Spatial coordinates, such as 2D points $(x, y)$ on the image plane (e.g., for end-effector trace prediction), are first normalized from their original pixel coordinates (where $0 \leq x < W$ and $0 \leq y < H$, with $W$ and $H$ being the image width and height, respectively) to a fixed integer range of $[0, 1024)$. The resulting integer pair is then formatted as a string (e.g., "(99,102)" as shown in Figure 4) and subsequently tokenized using the aforementioned Qwen2-VL text tokenizer. Bounding boxes, used for tasks like visual grounding, are handled in a similar manner by tokenizing their top-left $(x\_1, y\_1)$ and bottom-right $(x\_2, y\_2)$ corner points as text (e.g., "(105,114),(156,162)" in Figure 4). Sequences of point tokens are prefixed by the special token `<|point_start|>`, and bounding box token sequences are prefixed by `<|box_start|>`.

This comprehensive tokenization strategy, visually summarized in Figure 4, transforms complex, multimodal interaction sequences into a unified linear sequence of discrete tokens. The model can then be trained end-to-end using a standard cross-entropy loss objective for causal next-token prediction, irrespective of originating modality. This architectural unification simplifies the training paradigm and allows RoboOmni to effectively leverage powerful sequence modeling techniques across all aspects of the Vision-Language-Action task.

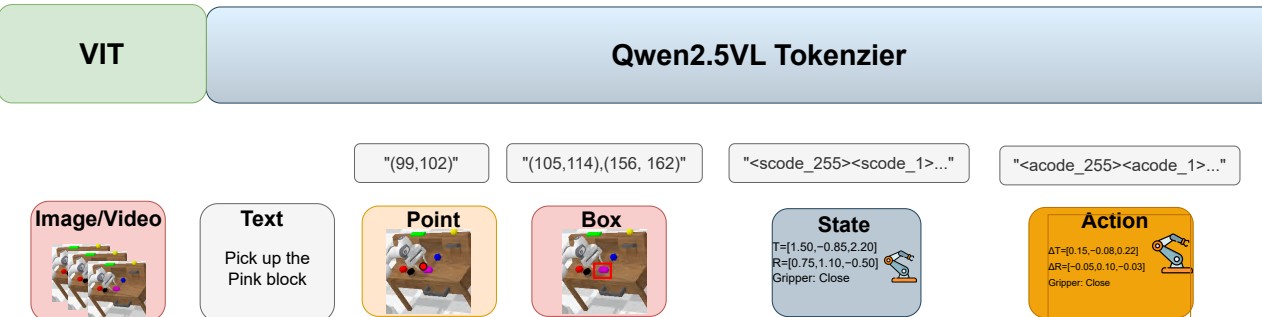

*Figure 4.* Overview of the unified modality tokenization pipeline in RoboOmni. Diverse inputs such as Image/Video, Text instructions, Point coordinates, Bounding Boxes (Box), Robot State, and Action commands are processed and converted into a shared discrete token sequence by the Qwen2.5-VL tokenizer. The figure shows schematic examples of raw inputs on the bottom row and their corresponding tokenized representations above them (e.g., point coordinates as text strings, states and actions as sequences of dedicated 'scode_X' and 'acode_X' tokens representing their discretized bin values).

## B. Multi-Modality Co-Training Data

A key advantage of our unified modality representation is the ability to seamlessly integrate auxiliary training tasks alongside direct action prediction. By co-training on a diverse set of objectives using the same next-token prediction framework, we enable the model to develop richer representations and transfer knowledge across modalities, ultimately benefiting the primary manipulation task and enhancing generalization (Team et al., 2025). Figure 5 illustrates examples of how multi-modal interleaved inputs are structured for various co-training tasks. The primary and auxiliary tasks, their data organization, sources, and the capabilities they impart to RoboOmni are detailed below.

### B.1. Action Prediction

The Action Prediction task is central to the function of RoboOmni as a Vision-Language-Action (VLA) model. Data for this task is organized into interleaved sequences representing timesteps of a robotic manipulation trajectory, following a format such as $V_1, L_1, S_1, A_1, V_2, L_2, S_2, A_2, \ldots, V_T, L_T, S_T, A_T$. Here, $V_t$ represents the visual observation (e.g., camera images), $L_t$ is the tokenized language instruction (which may be a constant task-level goal repeated across timesteps or a more dynamic input), $S_t$ is the proprioceptive state of the robot (e.g., joint angles, end-effector pose), and $A_t$ is the action executed at timestep $t$. The model is trained to predict the action tokens $A_t$ given the historical context of preceding vision, language, state, and action tokens. This data is primarily sourced from large-scale robotics datasets that provide expert demonstrations of manipulation tasks, including comprehensive collections like the Open X-Embodiment (OXE) dataset, as well as specific benchmarks such as Calvin (Mees et al., 2022b), RT-1 (Brohan et al., 2022), Droid (Khazatsky et al., 2024), and potentially custom-collected real-world robot interaction data. Training on this data endows Ro-

boOmni with the core capability to perform physical interactions and manipulations in its environment, effectively learning a policy that maps multimodal sensory inputs and language commands to sequences of robot actions required to complete specified tasks.

### B.2. Visual Question Answering (VQA)

For Visual Question Answering, the data is organized as triplets of (image, natural language question, natural language answer). The model receives an image and a question pertaining to its content and is trained to generate a concise and accurate textual answer. We utilize established VQA benchmarks for this objective, primarily the CLEVR dataset (Salewski et al., 2022) for its focus on compositional visual reasoning, and general VQA datasets like VQA v2 (Goyal et al., 2017) which cover a wider array of questions and visual concepts. Training on VQA preserves and enhances the core capabilities of the foundational Vision-Language Model (VLM) in sophisticated image understanding and nuanced text generation. This ensures RoboOmni retains strong multimodal reasoning skills crucial for interpreting complex instructions, analyzing scenes effectively, and potentially engaging in broader dialogue regarding its visual environment.

### B.3. Visual Grounding (Bounding Box Prediction)

In the Visual Grounding task, the model processes an image alongside a textual query or instruction that refers to one or more objects within that image, and it is trained to output the bounding box coordinates of the specified objects. These coordinates are discretized and then tokenized into a textual representation (as detailed in Appendix A), which the model predicts autoregressively. Data for this task is sourced from two main repositories: the COCO (Common Objects in Context) dataset (Chen et al., 2015), which provides extensive bounding box annotations for a wide variety of objects, and

**Multi-Modal Interleaved Input**

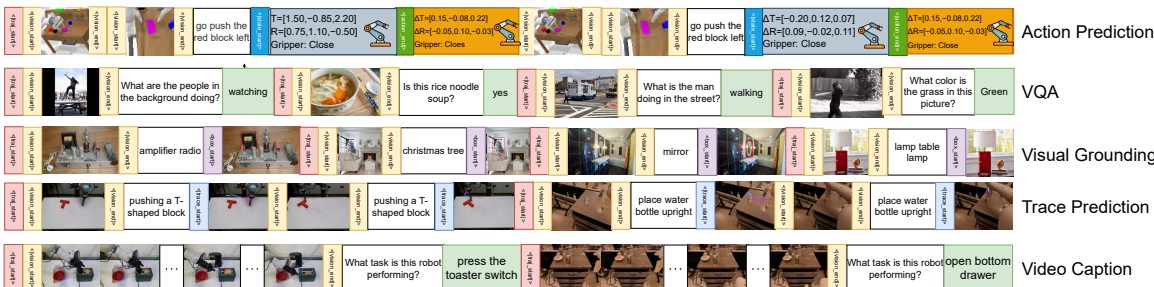

*Figure 5.* Examples of multi-modal interleaved input sequences for different co-training tasks within RoboOmni. The top row depicts a sequence for Action Prediction, incorporating visual input, language instruction, robot state, and the predicted action. Subsequent rows illustrate input formats for Visual Question Answering (VQA), Visual Grounding, and Trace Prediction, Video Caption, showcasing the diverse data types processed by the unified framework.

the blip3-grounding-50m dataset (Xue et al., 2024), specifically curated to enhance visual grounding capabilities. This training explicitly cultivates the spatial understanding of the model and object localization skills, which are critical for enabling precise robotic manipulation by allowing RoboOmni to accurately identify and locate objects relevant to the task or mentioned in instructions.

### B.4. Trace Prediction

The Trace Prediction task aims to instill an understanding of short-term motion dynamics and generalizable physical priors by training the model to predict 2D end-effector trajectories. Input 3D gripper coordinates are projected to 2D pixel points using camera parameters and then tokenized into a text-based representation (see Appendix A). Each trajectory is conceptualized as an interleaved sequence: $[l, o_1, \text{point}_1, o_2, \text{point}_2, \ldots, o_N, \text{point}_N]$, containing the language instruction $l$, initial visual observation $o_1$, and subsequent alternating visual observations $o_i$ with their corresponding 2D points $\text{point}_i$. During training, the sequence $S_{\text{train}}$ fed to the model is constructed by always preserving $l$ and $o_1$, but stochastically omitting subsequent visual observations $o_i$ (for $i > 1$) with a probability of 0.8, and similarly omitting point tokens $\text{point}_i$ with a probability of 0.2; the objective of the model is to predict the retained point tokens. Data for this task is drawn from the RLBench, Droid (Khazatsky et al., 2024), and Calvin (Mees et al., 2022b) datasets, offering diverse manipulation scenarios. This task, inspired by prior work (Team et al., 2025), enhances robustness and imputation skills due to the stochastic conditioning, while the 2D trace modality itself, offering a simplified and potentially cross-embodiment view of motion intent (Li et al., 2025), helps the model acquire broadly applicable physical principles.

### B.5. Video Captioning

For the Video Captioning task, data is structured as pairs of video segments (sequences of visual frames representing a history of observations) and their corresponding natural language descriptions or task summaries. These textual annotations serve as the prediction target given the video input. We primarily utilize videos paired with their task instructions from the Open X-Embodiment (OXE) dataset and the Calvin dataset (Mees et al., 2022b). Co-training on video captioning encourages RoboOmni to develop a deeper semantic understanding of complex interaction sequences over time. This enhances its ability to comprehend and follow language instructions by maintaining and refining its inherent language generation capabilities, ensuring a strong connection between dynamic visual information and its textual interpretation.

By jointly optimizing for these diverse objectives alongside the main action prediction task, RoboOmni learns more robust and generalizable representations. This multi-task co-training approach allows the model to leverage synergistic relationships between different modalities and tasks, leading to improved performance on the core robotic manipulation challenges and better adaptation to novel scenarios.

## C. Training Paradigm

This section details the core training methodologies employed in RoboOmni, emphasizing the rationale behind our choices and how they address common challenges in developing robust Vision-Language-Action (VLA) models. We explore alternatives and highlight the advantages of our selected approaches, particularly in facilitating a unified and efficient learning framework.

## C.1. Interleaved Input for Variable Vision and Action History

RoboOmni utilizes an interleaved data format to naturally incorporate variable-length historical context, comprising visual observations, language instructions, robot states, and past actions (e.g., $V_1, L_1, S_1, A_1, V_2, L_2, S_2, A_2, \ldots$). This approach inherently supports sequences of varying lengths, reflecting the dynamic nature of robotic tasks and interactions. A key aspect of this formulation is that the loss computation can incorporate signals from multiple action predictions within a single packed sequence, allowing for efficient learning from entire sub-trajectories. This method of conditioning on rich historical context for action decision-making aligns with recent advancements in large Vision-Language Models (VLMs) that also leverage extensive multimodal histories for improved understanding and generation, such as Emu (Wang et al., 2024). By adopting this paradigm, RoboOmni benefits from a natural and powerful way to model temporal dependencies and make informed, context-aware action choices.

## C.2. Classifier-Free Guidance for Vision-Motor Only Training

To enhance the robustness of the learned policies and enable training on trajectories that may lack explicit language annotations, we incorporate Classifier-Free Guidance (CFG) principles into our training regimen. During training, language instruction tokens are stochastically omitted from the input sequence with a predefined probability. This forces the model to learn to predict action sequences based solely on the visuomotor context (current and past visual observations and robot states). Such vision-motor only training helps the model to capture the inherent dynamics and continuity within action trajectories, independent of explicit language commands. Furthermore, this strategy allows us to leverage valuable demonstration data that may consist only of visual and state-action sequences, thereby broadening the effective training distribution and contributing to more generalizable and stable motor skills.

## C.3. Sequence Packing for Enhanced Training Efficiency and Multi-Distribution Learning

We employ sequence packing to improve GPU utilization and expose the model to a more diverse set of behavioral patterns within a single forward pass. Multiple independent sub-trajectories, potentially from different tasks or environments, are concatenated into a single long sequence, with appropriate padding and attention masking to prevent cross-contamination between distinct episodes. Unlike some sequence packing techniques in Large Language Models (LLMs) that might modify causal attention masks extensively to handle packed segments, our approach primarily relies on standard causal masking within each sub-trajectory, allowing the model to attend to all preceding tokens within its current episode. This design choice is partly inspired by findings, such as those in the DeepSeek-V3 technical report (Liu et al., 2024a), suggesting that simpler attention mechanisms in packed settings can yield strong performance. This method ensures that the computational benefits of Flash Attention mechanisms are maximally leveraged due to longer contiguous sequence processing. Moreover, training with packed sequences inherently promotes multi-distribution learning, as the model must infer the underlying task and dynamics from the immediate context within each packed segment. This capability is crucial for developing robust, context-dependent behaviors and lays a foundation for future work in few-shot adaptation, in-context learning (ICL), and chain-of-thought (CoT) reasoning within the robotics domain.

## C.4. Multi-Token Action Prediction (MTAP) for Action Chunking

Predicting a chunk of multiple future actions at each step, rather than a single action, can improve policy smoothness and planning horizon. However, implementing action chunking effectively within an autoregressive framework presents several challenges. A purely causal approach, where each action in a chunk is predicted sequentially, often suffers from compounding errors, as inaccuracies in earlier predicted actions negatively impact subsequent ones. Additionally, actions predicted earlier in the chunk cannot attend to information from actions that are supposed to occur later within the same chunk, limiting the coherence of the predicted action sequence. Alternative methods like the FAST tokenizer (Pertsch et al., 2025) attempt to address this by encoding the entire action chunk in the frequency domain, allowing temporal information across the chunk to be captured without causal limitations during encoding. However, during generation, actions are still typically decoded token by token (or dimension by dimension for each action in the chunk), which can lead to slower inference times for generating a complete action chunk. For instance, observations indicate that $\pi$-FAST requires significantly more time for generation (e.g., 750 ms) compared to models like $\pi_0$ (e.g., 100 ms) that generate chunks more directly (Black et al., 2024; Pertsch et al., 2025). Another strategy involves modifying the causal attention mask to allow tokens within an action chunk to attend to each other more freely, or even to allow all action tokens in a chunk to be predicted in parallel from a shared prefix. While this can enable fast, parallel generation of an action chunk, modifying the VLM's native causal attention structure can introduce complexities. Firstly, non-standard attention patterns can reduce the efficiency of attention computation mechanisms and, consequently, lower overall training throughput. Secondly,

and more critically for our framework, such modifications often make it difficult to support variable-length interleaved data formats that include multiple historical (vision, state, action) timesteps. Models adopting this approach, such as OpenVLA-OFT (Kim et al., 2024), often revert to using only a single frame of observation as input to the policy, thereby losing the benefits of historical context, which numerous studies have shown to be crucial for robust policy performance (Brohan et al., 2022; Li et al., 2023).

To address these limitations, RoboOmni employs Multi-Token Action Prediction (MTAP) (Gloeckle et al., 2024; Liu et al., 2024a). In MTAP, for predicting an action chunk of size $H$, the model processes the input history once through its shared transformer backbone. Then, instead of a single output head, $H$ parallel prediction heads (or a replicated final layer mechanism) are used, each dedicated to predicting one action step in the chunk. Specifically, from the final shared hidden state, $H$ distinct transformations are applied to produce $H$ sets of logits, one for each action $a_{t+k}$ where $k \in [0, H-1)$. MTAP offers several advantages. Firstly, this non-causal approach to predicting the action chunk avoids the issue of error accumulation inherent in sequential causal prediction. Unlike modifying the global causal mask, MTAP preserves the standard causal processing for the historical interleaved input sequence, allowing it to natively support rich vision-action history. Secondly, because MTAP involves generating a fixed number of output tokens (e.g., 7 tokens per action if each action has 7 dimensions) in parallel via multiple heads, regardless of the chunk length $H$, it is highly amenable to VLM infrastructure optimizations such as model parallelism and efficient batching. This allows for very fast generation of action chunks, potentially outperforming methods like FAST tokenizer in terms of speed, and remaining competitive with single-step generation models like $\pi_0$ or OFT-style approaches, even when incorporating extensive historical context. Thirdly, these parallel heads for action prediction do not interfere with the tokenization or prediction mechanisms for other modalities (text, vision features) or other co-training tasks (VQA, grounding), allowing RoboOmni to seamlessly benefit from diverse VL co-training. Finally, our experiments demonstrate that MTAP provides a significant performance uplift. The predicted action chunks can be effectively utilized with techniques such as receding horizon control and temporal ensembling to further enhance policy stability and task success rates.

### C.5. Model Backbone and Training Parameters

RoboOmni is built upon the Qwen2.5-VL-7B model as its foundational Vision-Language Model backbone. The original tokenizer of Qwen2.5-VL is expanded to include the necessary action tokens, state tokens, and special marker tokens as detailed in Appendix A. For training, we employ the AdamW optimizer with a learning rate of $1 \times 10^{-4}$. A cosine learning rate decay schedule is utilized, with a warm-up phase constituting 5% of the total training steps. A weight decay of 0.01 is applied to all trainable parameters to mitigate overfitting. The model is typically trained for a specified number of epochs depending on the dataset size and task complexity, with specific details provided in the main experimental sections of the paper. All training is conducted using mixed-precision (e.g., bfloat16) to optimize for speed and memory efficiency on modern GPU hardware.

## D. Simulation

This section outlines the configuration of the simulation benchmarks used for evaluating RoboOmni.

### D.1. CALVIN

CALVIN (Composable Action Language and Vision) (Mees et al., 2022b) serves as a benchmark for evaluating long-horizon, language-conditioned robotic manipulation policies. It features a simulated tabletop environment where a Franka Emika Panda arm performs a variety of tasks. The benchmark includes a dataset of approximately 24,000 human-teleoperated demonstrations, each annotated with natural language instructions. These demonstrations cover 34 distinct, predefined basic skills, such as "rotate blue block right," "move slider left," and "turn on light bulb." Trajectories in CALVIN are relatively short, typically under 64 timesteps each. The dataset is structured into four scene splits (A, B, C, and D), which allow for evaluating generalization to different visual and physical configurations. Our experiments utilize the ABCD splits for training. For evaluation, policies are typically required to complete a sequence of multiple consecutive tasks, and performance is measured by the success rates in achieving these sequential goals and the average number of tasks successfully completed per trial. Visual input is provided from both a static third-person camera and a wrist-mounted camera on the robot.

### D.2. Implementation Details

Our model, `RoboOmni`, is built upon the Qwen2.5-VL-7B backbone. We evaluate two versions based on the action tokenization scheme: `RoboOmni`(Bin) using a standard binning tokenizer, and `RoboOmni`(FAST) employing the FAST tokenizer. During training, we use a weighted data mixture with sampling weights of 0.8 for the standard CALVIN dataset, 0.2 for the CALVIN dataset prepared for CFG, and 1.0 for general VLM datasets. The model is trained for 18,000 steps (approximately 2 epochs on CALVIN data) with a global batch size of 64. We use a history length of 5, an action chunk size of 10, and pack sequences to a maximum length of 2048. For optimization, we use the AdamW optimizer with a weight decay of 0.1,

*Table 6.* Comprehensive experimental results and ablation studies on the CALVIN (ABCD→D) benchmark. This table aggregates all configurations evaluated in our study for a detailed comparison. The default configurations for **RoboOmni(Bin)** and **RoboOmni(FAST)** are highlighted in bold.

| Configuration | Top K Success Rate | | | | | Avg. Length |
|---|---|---|---|---|---|---|
| | Top 1 | Top 2 | Top 3 | Top 4 | Top 5 | |
| *Main Results: Baselines and Proposed Models* | | | | | | |
| OpenVLA | 0.921 | 0.732 | 0.565 | 0.455 | 0.346 | 3.03 |
| $\pi_0$-FAST (PaliGemma) | 0.974 | 0.936 | 0.892 | 0.848 | 0.803 | 4.45 |
| **RoboOmni(Bin) (Default)** | **0.997** | **0.973** | **0.940** | **0.895** | **0.834** | **4.64** |
| **RoboOmni(FAST) (Default)** | **0.997** | **0.982** | **0.951** | **0.918** | **0.881** | **4.73** |
| *Ablation: Without MTAP* | | | | | | |
| Tokenizer: BIN | 0.990 | 0.935 | 0.865 | 0.776 | 0.679 | 4.24 |
| Tokenizer: FAST | 0.990 | 0.961 | 0.909 | 0.860 | 0.801 | 4.52 |
| *Ablation: Bin Size (with MTAP)* | | | | | | |
| Bin Size = 128 (Tokenizer: BIN) | 0.989 | 0.955 | 0.920 | 0.890 | 0.837 | 4.59 |
| Bin Size = 1024 (Tokenizer: BIN) | 0.980 | 0.939 | 0.888 | 0.838 | 0.790 | 4.44 |
| Bin Size = 128 (Tokenizer: FAST) | 0.996 | 0.976 | 0.950 | 0.913 | 0.861 | 4.70 |
| Bin Size = 1024 (Tokenizer: FAST) | 0.990 | 0.968 | 0.940 | 0.916 | 0.871 | 4.68 |
| *Ablation: Window Size (Default: RoboOmni(Bin))* | | | | | | |
| Window Size = 1 | 0.973 | 0.932 | 0.897 | 0.871 | 0.813 | 4.49 |
| Window Size = 10 | 0.985 | 0.955 | 0.914 | 0.870 | 0.824 | 4.55 |
| *Ablation: Model Size (Default: RoboOmni(Bin))* | | | | | | |
| Qwen2-VL-2B | 0.981 | 0.939 | 0.886 | 0.842 | 0.776 | 4.42 |
| Qwen2.5-VL-3B | 0.984 | 0.952 | 0.911 | 0.875 | 0.819 | 4.54 |
| Qwen2-VL-7B | 0.982 | 0.956 | 0.918 | 0.881 | 0.828 | 4.57 |
| *Ablation: Training Strategies (Default: RoboOmni(Bin))* | | | | | | |
| Without VLM Dataset | 0.991 | 0.962 | 0.911 | 0.855 | 0.806 | 4.53 |
| Without Sequence Packing | 0.983 | 0.934 | 0.897 | 0.853 | 0.791 | 4.46 |
| Without CFG | 0.987 | 0.947 | 0.897 | 0.852 | 0.795 | 4.48 |

and employ a cosine learning rate schedule with a 1000-step warmup, a maximum learning rate of $1 \times 10^{-4}$, and a minimum of $1 \times 10^{-7}$.

### D.3. SimplerEnv (Google Robot) Implementation Details

For the Google Robot tasks evaluated in SimplerEnv, we utilize the same model architecture (RoboOmni based on Qwen2.5-VL-7B) and optimization strategy as detailed in Section D.2. Specifically, we employ the AdamW optimizer with a weight decay of 0.1 and a cosine learning rate schedule, featuring 1,000 warmup steps, a peak learning rate of $1 \times 10^{-4}$, and a minimum learning rate of $1 \times 10^{-7}$. Given the large scale of the Google Robot real-world dataset, which comprises approximately **3 million samples**, we train the model for **30,000 steps**.

We adapt the input configurations to the specific characteristics of the Google Robot domain. Visual observations are resized to a resolution of $224 \times 224$. To balance context

with computational efficiency in this setting, we utilize a history window size of 3 and predict action chunks of size 10. Furthermore, to maximize training throughput, we employ sequence packing, packing up to 4 independent trajectory samples into a single input sequence.

## E. Real Robot

To evaluate the performance of RoboOmni in the real world, we perform experiments on a real robot platform. The platform consists of a Kinova Gen-3 robot arm equipped with a Robotiq 2F-85 parallel-jaw gripper and two cameras, i.e., one static camera for capturing the workspace and another camera mounted on the end-effector. The training dataset consists of 18k human demonstrations across 37 tasks, which include 23 pick-and-place tasks and 14 non pick-and-place tasks such as pouring, flipping, and rotating.

We design four different settings to evaluate the model performance: Simple, Unseen Distractors, Unseen Instructions,

and Unseen Objects.

- In **Simple**, the scene is set to be similar to those in the training data.

- In **Unseen Distractors**, unseen distractors are added to the scene.

- In **Unseen Instructions**, we use GPT-4 to generate unseen synonyms for the verbs in the instructions. For example, we replace "pick up" with "take", "cap" with "cover", and "stack" with "pile".

- In **Unseen Objects**, the robot is instructed to manipulate objects that were not included in the training dataset. And the language instructions are adjusted accordingly, i.e., the language instructions are also unseen.

In total, we evaluate 30 different tasks: 18 of which were seen during training, while the rest were unseen. See the appendix on the project page for the full list of training tasks and the 30 evaluated tasks. We compare the performance of RoboOmni with OpenVLA (Kim et al., 2024), Octo (Team et al., 2024), GR-1 (Wu et al., 2023), RoboVLMS (Li et al., 2026), $\pi_0$-FAST (Pertsch et al., 2025).

**Generalization Capabilities**  RoboOmni demonstrates strong generalization to novel scenarios, a crucial attribute for practical robotic systems. In the "Unseen Objects" setting, where the robot was tasked with manipulating objects not encountered during training, RoboOmni achieved a success rate markedly superior to the compared baselines, as depicted in Figure 3. For instance, while manipulating entirely new objects, RoboOmni maintained a considerable level of performance, whereas other models exhibited a more pronounced degradation. The detailed task breakdown in Table 7 further corroborates this; specifically, Figure 3 shows RoboOmni (labeled as "Ours") achieving a success rate of **91.0%** in the aggregate "Unseen Objects" category. This substantially surpasses not only standard baselines but also strong concurrent methods like $\pi_0$-FAST (61.0%) and RoboVLMs (55.0%). This suggests that the unified modal representation and co-training strategies employed by RoboOmni contribute to a more abstract and transferable understanding of object properties and manipulation skills. Similarly, in the "Unseen Distractors" setting, RoboOmni maintained a high success rate (**89.0%**), significantly outperforming $\pi_0$-FAST (68.0%) when novel objects cluttered the scene. This indicates an exceptional ability to differentiate between target objects and irrelevant items.

**Instruction-Following Fidelity**  The ability to accurately interpret and execute commands based on varied linguistic inputs is paramount for Vision-Language-Action (VLA)

models. The performance of RoboOmni in the "Unseen Instructions" setting, where synonyms or paraphrased commands were provided (e.g., replacing "pick up" with "take", or "cap" with "cover"), highlights its robust language understanding. Figure 3 indicates that RoboOmni achieved a success rate of **89.0%** under "Unseen Instructions", again leading the compared models by a significant margin (compared to 63.0% for $\pi_0$-FAST and 50.0% for RoboVLMs). This level of performance suggests that RoboOmni is not merely memorizing command-action pairings but is developing a more nuanced semantic comprehension of the instructions. The high success rate in this category implies that the VLM backbone, enhanced by multi-modal co-training, effectively grounds novel linguistic expressions to corresponding robotic actions.

**Robustness**  Overall robustness is evaluated by the model's ability to consistently perform across a range of challenging, unseen conditions. RoboOmni consistently outperformed other models across all "Unseen" categories (Distractors, Instructions, Objects), and consequently, in the overall "Average" success rate shown in Figure 3 (**91.0%** for RoboOmni, compared to $\pi_0$-FAST 68.0%, RoboVLMs 60.0%, and GR-1 45.0%). Even in the "Simple" setting, designed to be similar to training data, RoboOmni achieved a dominant success rate (**93.0%**), establishing a strong baseline for precise control. The detailed Table 7 provides further evidence of this robustness. The consistent high performance, even when faced with novel objects, instructions, or distractors, underscores the stability and reliability of RoboOmni's learned policies. The Multi-Token Action Prediction (MTAP) strategy, combined with the comprehensive training paradigm including interleaved history and sequence packing, likely contributes to this enhanced robustness by enabling more coherent long-horizon reasoning and better adaptation to variations from the training distribution.

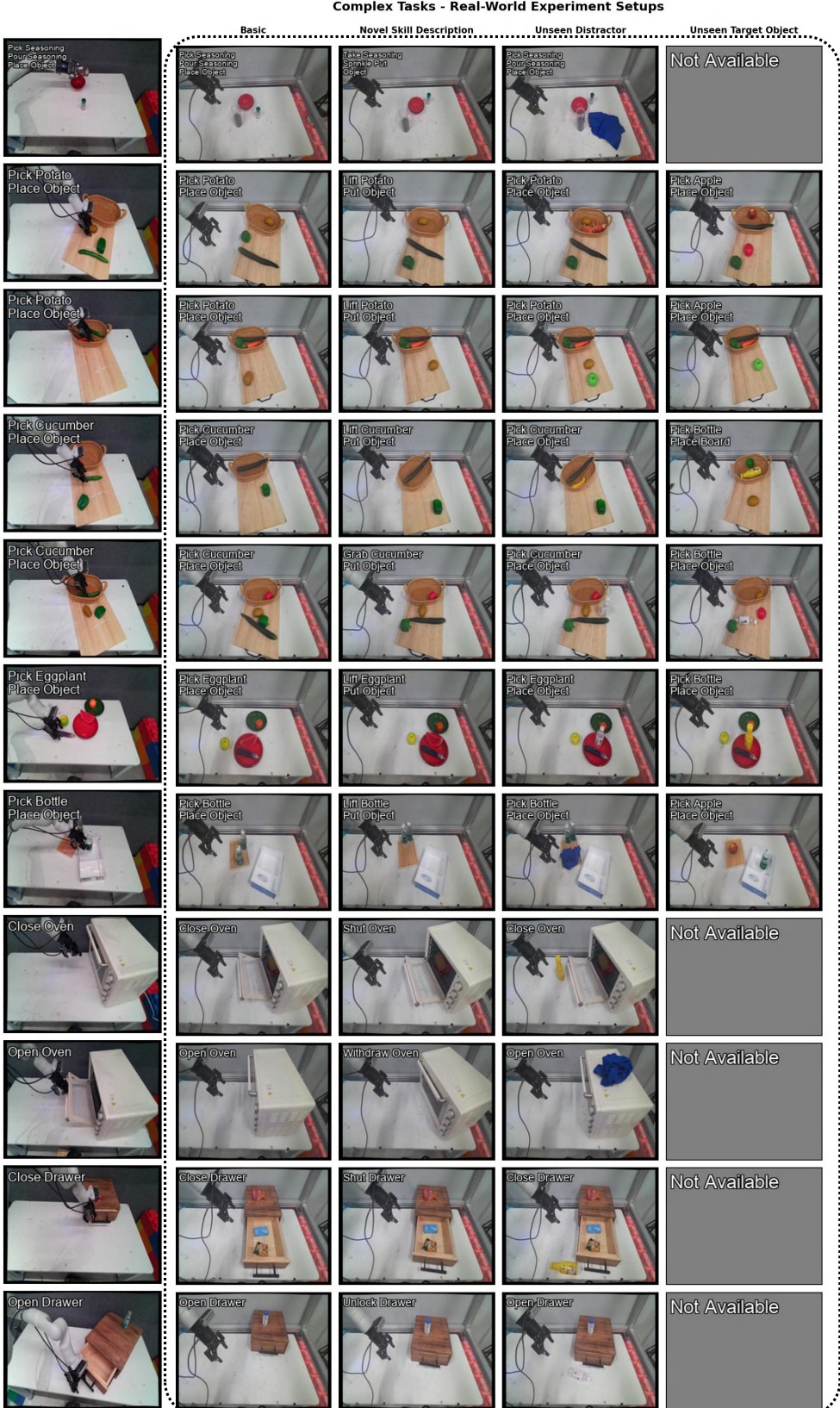

*Figure 6.* Real-world experimental setups for a variety of manipulation tasks. Each row illustrates a specific skill with a target object. The columns depict the same task under different experimental conditions.

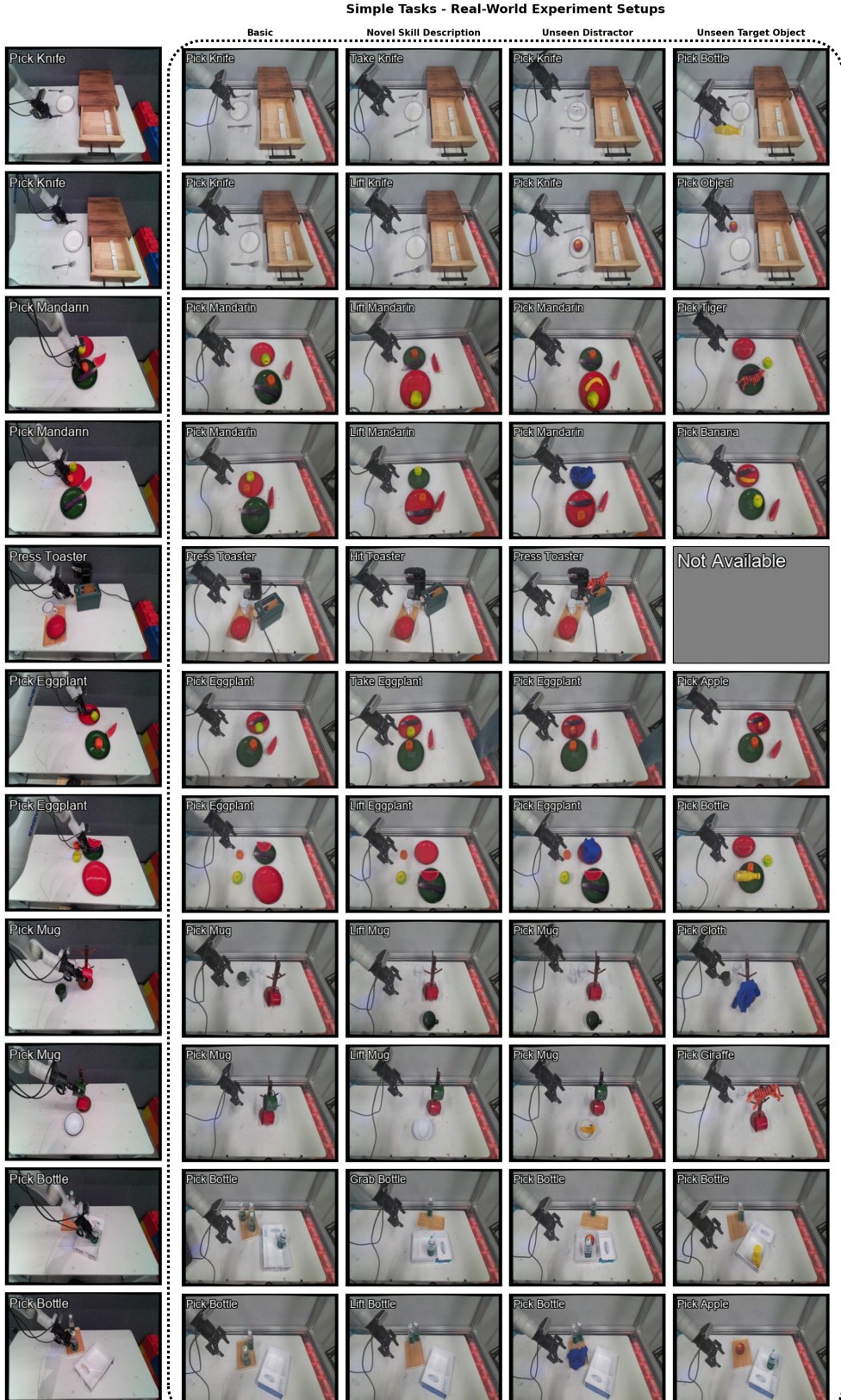

*Figure 7.* Real-world experimental setups for a variety of manipulation tasks. Each row illustrates a specific skill with a target object. The columns depict the same task under different experimental conditions.

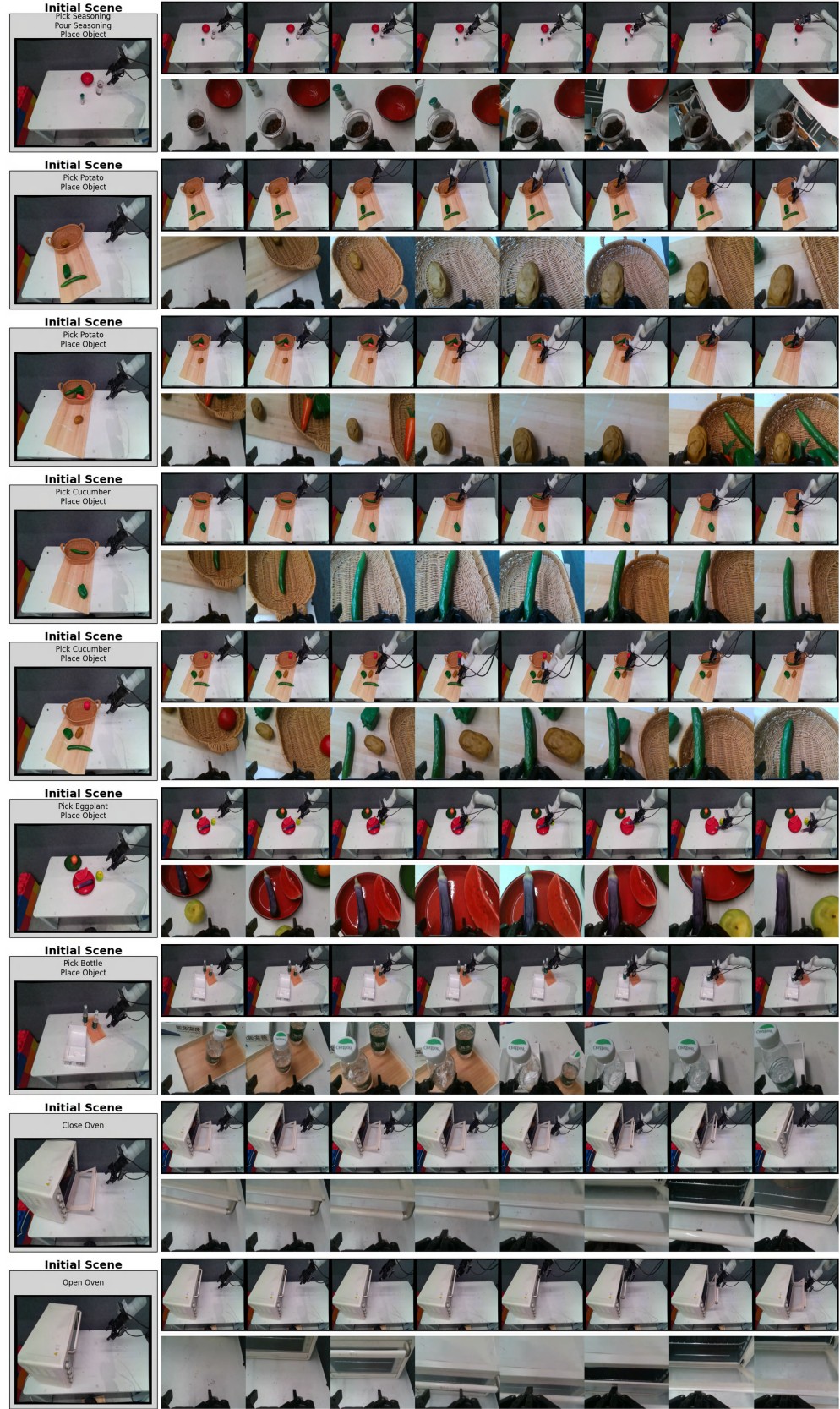

*Figure 8.* Qualitative results for basic setting.

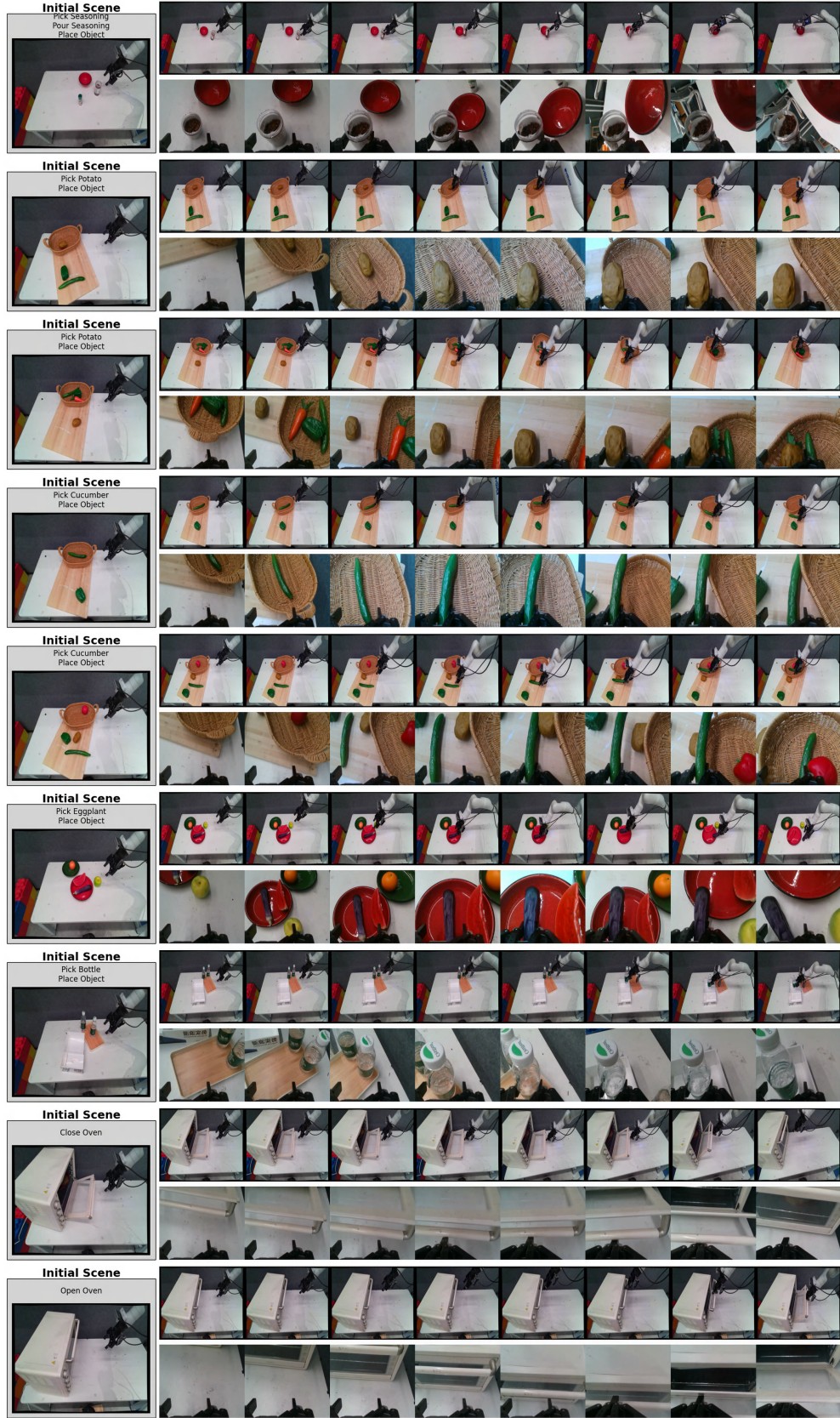

*Figure 9.* Qualitative results for unseen prompt setting.

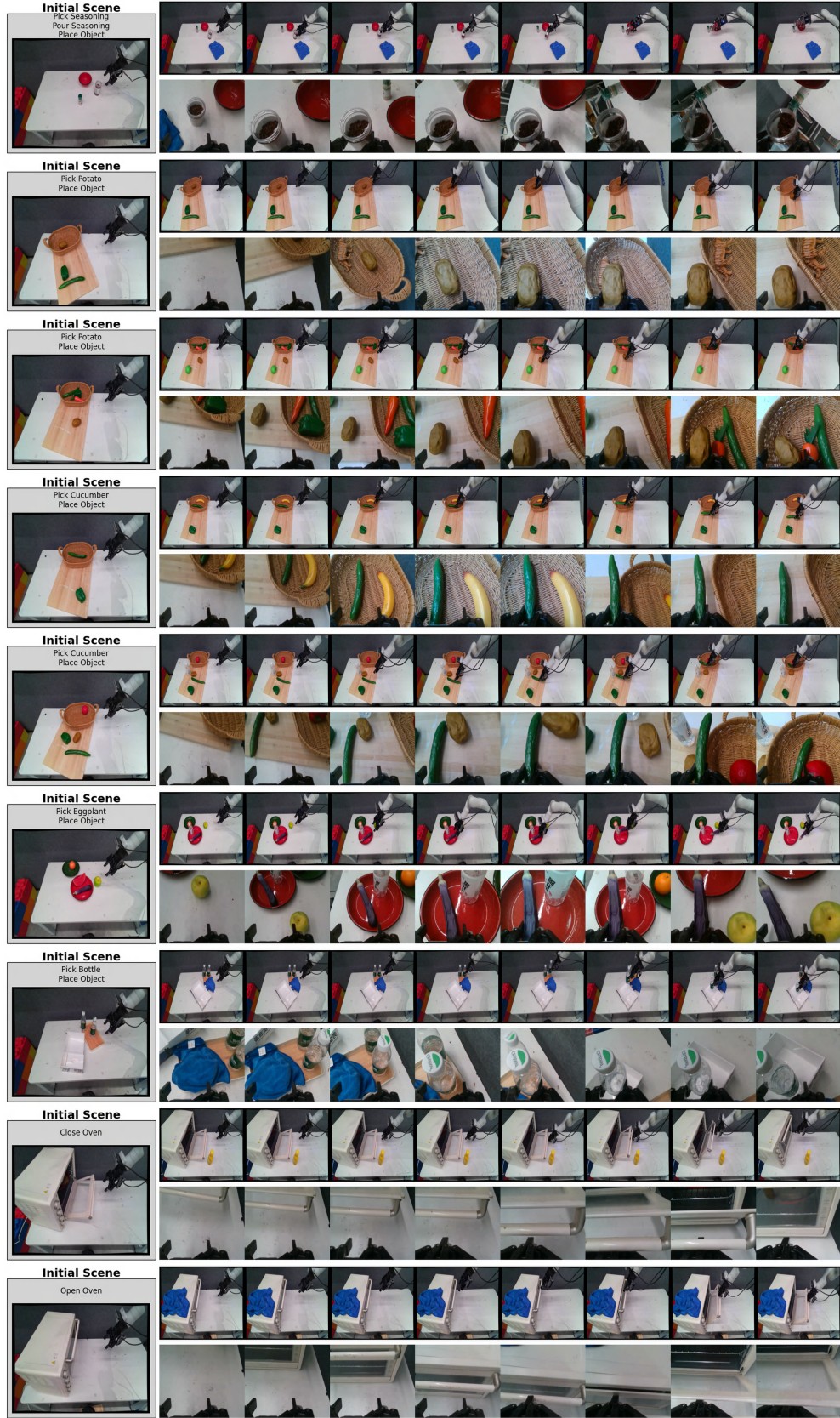

*Figure 10.* Qualitative results for unseen distractors setting

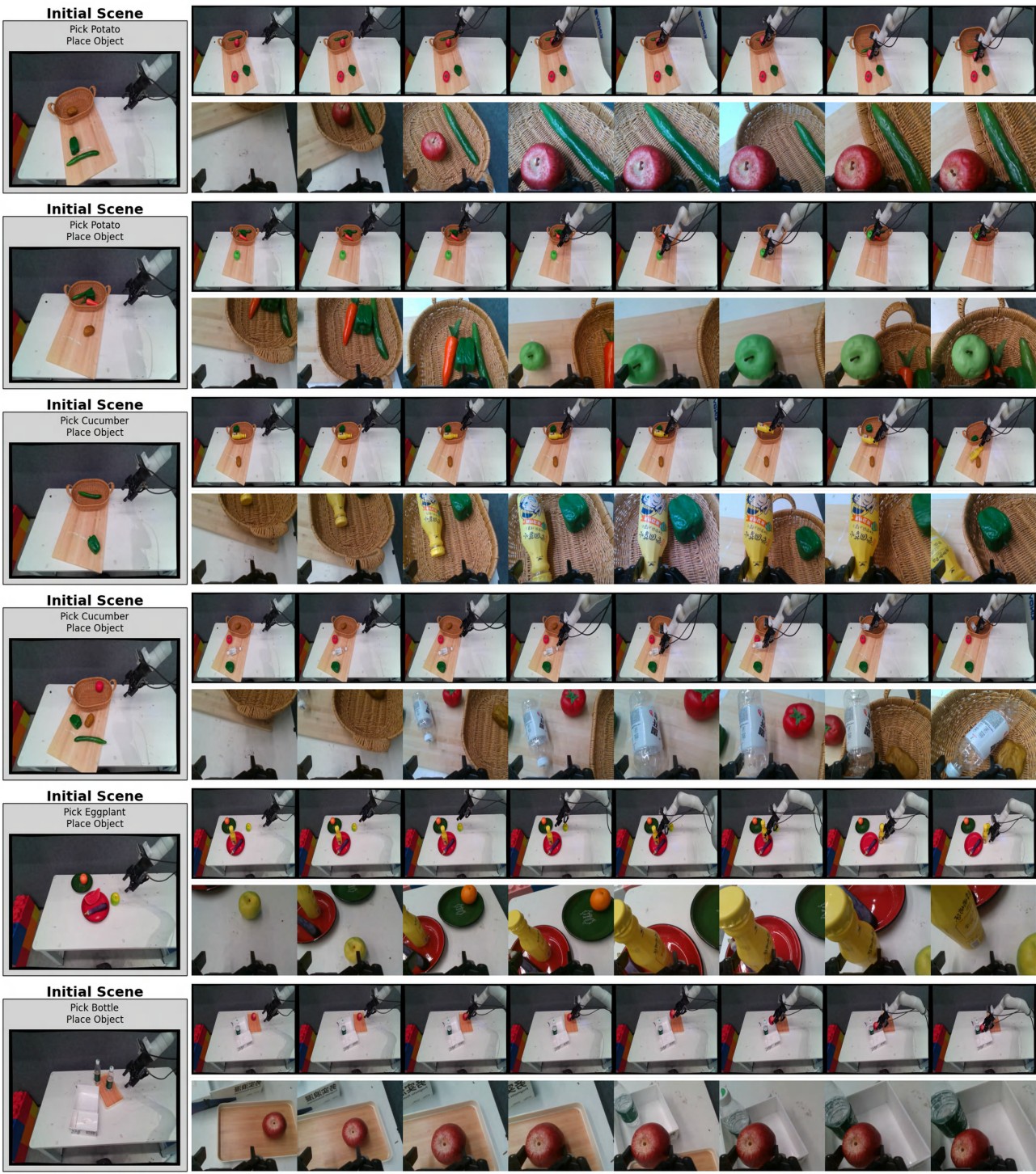

*Figure 11.* Qualitative results for unseen target object setting

*Table 7.* Detailed success rates (%) of RoboOmni across various real-world manipulation tasks and settings. The 'Basic' setting refers to the standard task setup. 'Prompt', 'Distractor', and 'Target Object' refer to settings with unseen prompts, unseen distractors, and unseen target objects, respectively.

| Task | Basic | Prompt | Distractor | Target Object |
|---|---|---|---|---|
| pour the black seasoning powder in the red bowl | 100.0 | 83.3 | 50.0 | N/A |
| press the toaster switch | 100.0 | 100.0 | 100.0 | N/A |
| close the drawer | 100.0 | 100.0 | 100.0 | N/A |
| open the drawer | 83.3 | 50.0 | 50.0 | N/A |
| close the oven | 100.0 | 83.3 | 100.0 | N/A |
| open the oven | 83.3 | 83.3 | 33.3 | N/A |
| pick up the cucumber from the vegetable basket; place the picked object on the cutting board | 100.0 | 66.7 | 100.0 | 66.7 |
| pick up the cucumber from the cutting board; place the picked object in the vegetable basket | 83.3 | 83.3 | 100.0 | 100.0 |
| pick up the potato from the vegetable basket; place the picked object on the cutting board | 50.0 | 83.3 | 66.7 | 83.3 |
| pick up the potato from the cutting board; place the picked object in the vegetable basket | 100.0 | 66.7 | 83.3 | 83.3 |
| pick up the eggplant from the red plate; place the picked object on the table | 100.0 | 100.0 | 100.0 | 100.0 |
| pick up the green bottle from the tray; place the picked object in the white box | 100.0 | 100.0 | 100.0 | 100.0 |
| pick up the knife from the right of the white plate | 83.3 | 100.0 | 100.0 | 100.0 |
| pick up the knife from the left of the white plate | 83.3 | 83.3 | 83.3 | 83.3 |
| pick up the eggplant from the red plate | 100.0 | 100.0 | 100.0 | 100.0 |
| pick up the eggplant from the green plate | 100.0 | 100.0 | 100.0 | 100.0 |
| pick up the mandarin from the green plate | 100.0 | 100.0 | 100.0 | 100.0 |
| pick up the mandarin from the red plate | 100.0 | 100.0 | 100.0 | 83.3 |
| pick up the red mug from the rack | 100.0 | 83.3 | 100.0 | 100.0 |
| pick up the green mug from the rack | 100.0 | 100.0 | 100.0 | 100.0 |
| pick up the green bottle from the white box | 100.0 | 100.0 | 100.0 | 100.0 |
| pick up the green bottle from the tray | 83.3 | 100.0 | 100.0 | 100.0 |

