# OpenReview forum: "RoboOmni: Actions Are Just Another Modality for Vision-Language Models"
_ICML.cc/2026/Conference — ICML 2026 regular_

### Official Review · Reviewer_WvzF · 2026-03-09

**Soundness:** 3
**Presentation:** 4
**Significance:** 3
**Originality:** 3
**Overall Recommendation:** 3
**Confidence:** 3

**Summary:**

This paper studies unified Vision-Language-Action (VLA) modeling by treating robot actions as another modality within a next-token prediction framework. This study attempts to focus on a central concept: integrating vision, language, state, and action tokens into a single autoregressive transformer architecture.
The authors propose RoboOmni, a multimodal framework that interleaves visual observations, language instructions, robot states, and actions into a unified token sequence. The key technical component is Multi-Token Action Prediction (MTAP), which enables parallel prediction of multiple future actions and mitigates compounding errors associated with sequential action decoding. The framework preserves the training and inference pipeline of standard VLMs and supports multimodal co-training tasks such as visual grounding, trace prediction, and VQA.
The model is evaluated on the CALVIN benchmark, SimplerEnv real-to-sim tasks, and real robot experiments. The results show competitive performance compared with several unified and decoupled VLA baselines. Overall, this paper assesses the concept of modeling robot control within a unified multimodal token prediction paradigm.

**Compliance With Llm Reviewing Policy:**

Affirmed.

**Key Questions For Authors:**

1.	How sensitive is RoboOmni to the action discretization scheme, particularly for tasks requiring very fine-grained continuous control? The bin size ablation (Table 4) provides some insight, but it would be valuable to see experiments on tasks with inherently continuous and high-precision requirements (e.g., contact-rich assembly or deformable object manipulation).
2.	The improvements may result from multiple design choices (MTAP, multimodal co-training, larger backbone). Could the authors clarify the relative contribution of these components? In particular, how much of the gain over baselines like π0-FAST is attributable to using a stronger Qwen2.5-VL-7B backbone versus the architectural innovations?
3.	Does the cross-trajectory sequence packing strategy introduce unintended dependencies between trajectories during training? The paper mentions intentionally omitting attention masks between packed samples, but this design choice lacks sufficient empirical justification. An ablation comparing masked vs. unmasked packing would strengthen the claim.
4.	Given the rapid progress in the VLA field, how does RoboOmni compare with more recent models such as π0.5, OpenVLA-OFT, and other concurrent works? The current baselines (OpenVLA, RoboFlamingo, GR-1, Octo) are relatively dated, and the claim of state-of-the-art performance would be significantly strengthened by comparisons with these newer and stronger methods. If direct experimental comparison is not feasible, could the authors provide a detailed qualitative or theoretical analysis of the expected relative performance?

**Limitations:**

Yes. The authors discuss potential dataset bias and safety risks when deploying robotic manipulation models in real-world environments. However, additional discussion on the scalability limitations of discrete tokenization for higher-DOF systems and multi-robot coordination would further strengthen the paper.

**Strengths And Weaknesses:**

Strengths
1.	Clear and simple unified formulation. The paper presents a clean architecture that treats actions as another modality in the multimodal token sequence, which simplifies the integration between VLM reasoning and robot control.
2.	Effective action chunking mechanism. The proposed MTAP module enables parallel prediction of action chunks and reduces inference latency compared with standard autoregressive decoding.
3.	Strong empirical evaluation across multiple settings. The paper reports experiments across simulation benchmarks (CALVIN), real-to-sim evaluation (SimplerEnv), and real robot experiments, with consistent improvements over the compared prior methods.
4.	Practical inference efficiency. The unified framework maintains compatibility with modern VLM optimizations such as KV caching, leading to significant speed improvements.
Weaknesses
1.	Moderate novelty of the core idea. The idea of representing robot actions as tokens in a multimodal transformer has been explored in prior VLA models (e.g., RT-2, OpenVLA). The primary novelty mainly lies in the MTAP mechanism and training strategy.
2.	Limited analysis of discrete action modeling. While the paper argues that discrete tokenization can outperform continuous policies, there is limited discussion of scenarios where discretization may fail (e.g., highly dynamic or precision-sensitive control).
3.	Performance gains are not fully attributed. The improvements may arise from multiple factors, including MTAP, multimodal co-training, and backbone scaling. The paper could provide clearer analysis of which component contributes most to the gains.
4.	Limited discussion of potential failure cases. The paper focuses on success rates but does not provide qualitative examples or analysis of failure modes.
5.	Outdated baselines and insufficient comparison with recent state-of-the-art. The experimental comparisons primarily rely on relatively older VLA models such as OpenVLA, RoboFlamingo, GR-1, and Octo, most of which were published before mid-2024. However, given the rapid progress in the VLA field, several stronger and more recent baselines have emerged by the time of this submission, including π0.5 (Physical Intelligence, 2025), OpenVLA-OFT, SpatialVLA (which is compared on SimplerEnv but absent from CALVIN and real robot evaluations), and HPT, among others. The paper mentions π0.5 and Gemini Robotics in the related work section but does not include them in the experimental evaluation. Without direct comparisons against these more recent models, the claim of achieving state-of-the-art performance is not fully convincing. The strong results on CALVIN and SimplerEnv may partly reflect the use of a more powerful backbone (Qwen2.5-VL-7B) and more training data rather than the proposed architectural innovations alone. The authors should either include comparisons with these recent baselines or provide a thorough discussion explaining why such comparisons are not feasible and how the proposed method is expected to perform relative to them.

---

> ### Author Rebuttal · Authors · 2026-03-31
>
> We thank the reviewer for the constructive feedback.
>
> **W1 (novelty).** We agree that action-as-token itself is not new. Our contribution is MTAP: for Bin it predicts an action horizon as temporal compression, while for FAST it acts as an auxiliary objective for frequency-domain tokens. The main claim is therefore not "first action tokenization," but that this MTAP-based unified discrete design makes a discrete VLA competitive enough to outperform prior discrete models and strong continuous baselines on CALVIN. In other words, the novelty lies in the specific MTAP mechanism and its integration into a unified VLM pipeline, not in discrete action tokens alone. This distinction matters because prior discrete VLAs already tokenized actions yet still lagged stronger continuous policies; our point is that MTAP changes that competitiveness rather than merely rephrasing the action representation.
>
> **W2/Q1 (limits of discretization).** Table 4 in the paper shows that 256 bins gives the best learnability-precision tradeoff; its normalized resolution is about 0.004 per dimension. To test a harder setting beyond CALVIN/SimplerEnv, we evaluated RoboCasa-Kitchen with chunk size 32 and obtained 68.2% average success, outperforming π0 (62.5%) and GR00T-N1.5 (64.1%). This suggests that discrete action modeling remains effective for longer horizons, articulated-object tasks, and richer scene diversity. At the same time, we agree that highly dexterous, force-sensitive, contact-rich, or very high-DoF settings remain important limitations and should be discussed explicitly. Our rebuttal is therefore not that discretization has no limits, but that its useful regime is broader than suggested by the original benchmark set.
>
> | Method | # Demos/Task | Avg SR (%) |
> |---|---|---|
> | π0 | 300 | 62.5 |
> | GR00T-N1.5 | 300 | 64.1 |
> | **RoboOmni(FAST)** | **300** | **68.2** |
>
> **W3/Q2 (attribution).** Backbone scaling alone does not explain the gains. We performed same-backbone and cross-backbone controls:
>
> | Config | Backbone | Top-1 | Top-5 | Avg Len |
> |---|---|---|---|---|
> | π0-FAST (orig.) | PaliGemma-3B | 97.4 | 80.3 | 4.45 |
> | π0-FAST (re-impl.) | Qwen2.5-VL-7B | 99.4 | 76.9 | 4.44 |
> | RoboOmni(FAST) | Qwen2.5-VL-7B | 99.7 | 88.1 | 4.73 |
> | RoboOmni(FAST) | Qwen3-VL-4B | 100.0 | 87.0 | 4.72 |
>
> The same-backbone comparison shows a large gap between π0-FAST and RoboOmni on Qwen2.5-VL-7B (76.9 vs. 88.1 Top-5), while the cross-backbone comparison for RoboOmni is very stable (88.1 vs. 87.0). Put differently, simply swapping PaliGemma for Qwen does not improve π0-FAST, whereas adding MTAP and the RoboOmni training design yields a large gain on the same backbone. MTAP itself adds +8.0 Top-5 for FAST and +15.5 for Bin (Table 3 in the paper), while VLM co-training, packing, and CFG contribute +2.8, +4.3, and +3.9, respectively (Table 5). Together, these results suggest that the main gains come from MTAP and training strategy rather than backbone scaling alone.
>
> **W4 (failure cases).** We observe two main failure modes: (1) chunk-tail inaccuracies, such as premature grasping or overshooting, consistent with the Task-1 to Task-5 drop on CALVIN; and (2) collisions in cluttered scenes due to the lack of force/tactile feedback and explicit 3D reasoning. In real robot rollouts, the first mode appears as early gripper closure or over-travel near the end of a predicted chunk, while the second appears as unintended contact with nearby objects or obstacles. We will add representative examples in the revision to make these failure modes explicit rather than reporting only aggregate success rates.
>
> **W5/Q4 (recent baselines).** We reproduced the requested recent baselines on CALVIN ABC→D under a unified protocol using official code when available and best-effort re-implementations otherwise:
>
> | Method | Top-1 | Top-5 | Avg Len |
> |---|---|---|---|
> | π0 | 93.8 | 59.9 | 3.84 |
> | π0.5 | 94.8 | 64.3 | 3.97 |
> | GR00T N1 | 94.2 | 66.8 | 4.01 |
> | π0-FAST | 98.9 | 69.8 | 4.24 |
> | OpenVLA-OFT | 95.7 | 70.7 | 4.12 |
> | RoboOmni(Bin) | 98.8 | 72.1 | 4.30 |
> | **RoboOmni(FAST)** | **99.2** | **73.5** | **4.35** |
>
> RoboOmni(FAST) leads all reproduced baselines, including +13.6 over π0, +9.2 over π0.5, +6.7 over GR00T N1, and +2.8 over OpenVLA-OFT in Top-5. These results directly address the reviewer's concern that our original baseline set was outdated, especially for π0.5 and OpenVLA-OFT, which were explicitly requested. SpatialVLA is already compared on SimplerEnv in our paper (+16.8 for RoboOmni), while HPT does not report CALVIN or SimplerEnv results.
>
> **Q3 (cross-trajectory packing).** We ran the requested direct ablation on CALVIN ABCD→D with MTAP+FAST:
>
> | Packing | Top-1 | Top-5 | Avg Len |
> |---|---|---|---|
> | Packed, unmasked | **99.7** | **88.1** | **4.73** |
> | Packed, masked | 99.5 | 85.5 | 4.66 |
>
> Unmasked packing is +2.6 Top-5 and +0.07 Avg-Len better, indicating that cross-trajectory visibility is beneficial in this setting.

---

> > ### Author Rebuttal · Reviewer_WvzF · 2026-04-02
> >
> > I thank the authors for the thorough rebuttal. The additional experiments address most of my concerns. One point remains: the recent baseline comparisons (π0.5, OpenVLA-OFT, GR00T N1) rely on re-implementations rather than official reported numbers, and differences in training setup could affect the fairness of comparison.
> > I acknowledge the rebuttal and maintain my current score.

---

> > > ### Author Response · Authors · 2026-04-03
> > >
> > > We thank the reviewer for the follow-up and for clarifying the remaining concern.
> > >
> > > We agree that these are not official CALVIN ABC→D results and should therefore be interpreted as controlled reproductions rather than official benchmark numbers.
> > >
> > > That said, we believe the comparisons are still informative for two reasons.
> > >
> > > First, no official CALVIN ABC→D results exist for π0, π0.5, GR00T N1, or OpenVLA-OFT. π0.5 reports only real-world mobile manipulation, GR00T N1 reports RoboCasa and DexMimicGen, OpenVLA-OFT reports LIBERO and ALOHA, and π0 does not report CALVIN results. As a result, a direct comparison on CALVIN necessarily requires re-running these methods under a shared protocol, using official code when available and best-effort re-implementations otherwise.
> > >
> > > Second, our reproduced numbers are externally calibrated rather than isolated. Most importantly, VLA-Adapter [1] independently reproduces OpenVLA-OFT on CALVIN ABC→D and reports Avg Len = 4.10, which is extremely close to our 4.12. Because this agreement comes from a separate research group using a different codebase, we view it as the strongest evidence that our evaluation is reasonably calibrated rather than favorable to our method.
> > >
> > > For the other reproduced baselines, the numbers also fall in a plausible range relative to published CALVIN results from related methods: our π0 result (3.84) lies between UniVLA-early (3.80) [2] and Seer (3.98) [1], and our GR00T N1 result (4.01) is in line with MoDE (4.01) [1] and LLaDA-VLA (4.01) [3]. We see these comparisons as sanity checks, not as substitutes for official results.
> > >
> > > We will make this point explicit in the revision: these recent baselines are controlled reproductions under a shared CALVIN protocol, not official reported numbers, and should be interpreted accordingly. Beyond them, Table 1 in the paper already compares RoboOmni against methods with official CALVIN results (π0-FAST, RoboVLMs, OpenVLA, GR-1), where RoboOmni remains stronger under the same training and evaluation protocol.
> > >
> > > We hope this clarifies the remaining concern, and we would be happy to share our training configurations and evaluation code for verification.
> > >
> > > **References:**
> > >
> > > [1] Wang Y, Ding P, Li L, et al. VLA-Adapter: An Effective Paradigm for Tiny-Scale Vision-Language-Action Model[C]//Proceedings of the AAAI Conference on Artificial Intelligence. 2026, 40(22): 18638-18646.
> > >
> > > [2] Wang Y, Li X, Wang W, et al. Unified Vision-Language-Action Model[J]. arXiv preprint arXiv:2506.19850, 2025.
> > >
> > > [3] Wen Y, Li H, Gu K, et al. LLaDA-VLA: Vision Language Diffusion Action Models[J]. arXiv preprint arXiv:2509.06932, 2025.

---

### Official Review · Reviewer_yUN2 · 2026-03-10

**Soundness:** 3
**Presentation:** 4
**Significance:** 3
**Originality:** 3
**Overall Recommendation:** 4
**Confidence:** 3

**Summary:**

This paper presents RoboOmni, a unified multimodal next-token prediction framework that innovatively treats actions as an integral modality within vision-language models (VLMs). RoboOmni enables full reuse of VLM-native training paradigms, including co-training with diverse multimodal data and inference pipelines, thereby achieving highly efficient model training and inference. And the exe results show that this paradigm performs very well

**Compliance With Llm Reviewing Policy:**

Affirmed.

**Final Justification:**

After rebuttal, my concerns have been resolved,so I maintain my positive assesment.

**Key Questions For Authors:**

See weakness

**Limitations:**

yes

**Strengths And Weaknesses:**

## Strength
1. Successfully integrates actions into the unified prediction paradigm, simplifying the architectural complexity of VLA (Vision-Language-Action) models.
2. RoboOmni enables full reuse of VLM-native training paradigms and is able to leverage the progress in VLMs in both training and inference pipelines
3. The paper covers a wide range of benchmarks as well as real-world deployment validations, with detailed data and reasonably chosen baseline comparisons. The model is robust when facing unseen tasks or objects, which is crucial for open-world robotic applications. which is crucial  for open-world robotic applications.
## Weakness
1. Although discrete tokenization simplifies modeling, autoregressive sampling may result in higher inference latency when generating long action sequences. The paper lacks specific quantitative analysis of real-time control frequency
2. Discretization may still result in loss of accuracy when handling extremely fine tasks that require real-time force feedback adjustments (such as inserting and removing tiny electronic components), and the article provides relatively little discussion on the limitations of such extreme situations.
3. The model has not been cross-validated on robots with different hardware architectures. If it is trained only on a specific robotic arm, how does its 'action-as-modality' generalize across hardware with different dynamic characteristics?

---

> ### Author Rebuttal · Authors · 2026-03-31
>
> We thank reviewer for the supportive review and thoughtful suggestions. We address each concern below.
>
> ## Response to W1: Inference Latency Analysis
>
> We appreciate this concern. Inference latency is already reported in Table 3 of the paper. We summarize the key results here and provide additional context:
>
> | Configuration | Tokens to Decode | Decoder Latency (ms/action) |
> |---|---|---|
> | MTAP + BIN | 7 tokens (one per DoF) | 12.1 ms |
> | MTAP + FAST | ~20 tokens (avg) | 17.5 ms |
> | w/o MTAP + BIN | 7 × H tokens (sequential) | 107 ms |
>
> The key point is that MTAP with Bin tokenization requires decoding only **7 tokens** (one per DoF) to generate the entire action chunk in parallel, whereas FAST tokenization requires about **20 tokens** per chunk on average. With SGLang (Zheng et al., 2024a) providing optimized serving, these decoding latencies are consistent with typical manipulation control frequencies (10-30 Hz). Without MTAP, however, the Bin tokenizer must decode 7 x H tokens sequentially, which increases latency to 107 ms/action and could hinder reactive control.
>
> We will add an explicit control frequency analysis in the revised paper to make this more prominent.
>
> ## Response to W2: Limitations for Extremely Fine Tasks
>
> We agree that this point deserves more discussion. The precision of discrete action modeling is bounded by the number of bins: with 256 bins over a normalized action space and 7-DoF control, the resolution in each dimension is approximately 1/256 (~0.004). This resolution is sufficient for the manipulation tasks evaluated in our work. Table 4 compares bin sizes of 128, 256, and 1024, and shows that 256 bins provides the best trade-off between learnability and precision.
>
> However, we acknowledge that tasks requiring **real-time force-feedback adjustments** (e.g., delicate electronic component insertion or deformable object manipulation) pose challenges that go beyond discretization precision. Such tasks fundamentally require closed-loop reactive control based on tactile or force sensing, so the limitation is not solely a consequence of the action representation. Incorporating tactile and force feedback into the observation space is therefore a promising direction for future work and would naturally complement RoboOmni's discrete action framework.
>
> We will expand the limitations discussion in the revised paper to explicitly address this.
>
> ## Response to W3: Cross-Hardware Generalization
>
> We acknowledge that cross-embodiment generalization with a single model has not been validated in this work, and we do not claim this capability in the paper. Each evaluation setting (CALVIN, SimplerEnv, Real Robot) uses a model trained specifically for that embodiment.
>
> That said, our current results do demonstrate strong **within-embodiment generalization** on the real-robot platform:
>
> | Setting | RoboOmni | π0-FAST |
> |---|---|---|
> | Unseen Distractors | 89% | 68% |
> | Unseen Instructions | 89% | 63% |
> | Unseen Objects | 91% | 61% |
>
> RoboOmni achieves 91% success on unseen objects (+30% over π0-FAST), showing that the "action-as-modality" framework generalizes well to novel visual and linguistic conditions within a given embodiment.
>
> Cross-embodiment generalization, i.e., training a single model to control robots with different dynamics, is an important and active research direction. In future work, we plan to investigate whether RoboOmni's unified tokenization framework can represent actions from different embodiments within a shared discrete vocabulary.

---

> > ### Author Rebuttal · Reviewer_yUN2 · 2026-04-02
> >
> > After rebuttal, my concerns have been resolved,so I maintain my positive assesment.

---

### Official Review · Reviewer_ivp2 · 2026-03-12

**Soundness:** 2
**Presentation:** 3
**Significance:** 2
**Originality:** 2
**Overall Recommendation:** 4
**Confidence:** 4

**Summary:**

This paper proposes a unified training framework that integrates multiple modalities by interleaving language, vision, and action tokens. The proposed method adopts a multi-token prediction scheme to reduce errors that occur during the action chunk generation process.

**Compliance With Llm Reviewing Policy:**

Affirmed.

**Final Justification:**

The authors have addressed most of my concerns. The remaining concern regarding the baseline was clarified during the discussion. Therefore, I increase my score to 4.

**Key Questions For Authors:**

Interleaving different modalities for joint training could potentially provide additional benefits, such as improved generalization. Have the authors observed any empirical improvements arising from this design?

**Limitations:**

yes

**Strengths And Weaknesses:**

Strengths
1. The absolute performance of the model is quite impressive, showing consistent improvements across Simpler, CALVIN, and real-world setups.

2.The concept of predicting action chunks through multi token prediction is interesting. Since, chunk prediction was typically performed through single parallel prediction (openvla-oft).


Weaknesses
1. Compressing long action chunks may become vulnerable when performing dexterous tasks or when the action frequency or degrees of freedom (DoF) increase. Meanwhile, the tasks presented in the paper (Simpler and CALVIN) are relatively simple, and based on the current experiments, it remains unclear whether the proposed method would work well in more dexterous environments. Evaluations on more complex settings such as RoboCasa-Kitchen or high-DoF setups like RoboTwin2.0 seem particularly necessary.

2. It is already known that robot-related VQA training can improve VLA performance. Similarly, chunk-level action prediction (e.g., OpenVLA-OFT), interleaved multimodal training (e.g., Octo), and multi-token prediction (widely studied in the LLM literature) are also existing concepts. Conceptually, each of these elements does not appear fundamentally new.

3. The paper claims modality integration based on discrete tokens; however, this also applies to approaches that use action tokenizers such as pi0-FAST. This aspect may contain some degree of overclaim.

4. Lack of baselines.
- The technically new aspect of the paper is that it adopts a different approach from parallel decoding when performing multi-token prediction. However, comparisons with relevant baselines are limited. Comparisons with methods similar to OpenVLA-OFT or related approaches seem necessary.
- Although the paper proposes multi-token prediction, there is a lack of baselines where the model is trained naively with Bin or FAST tokenizers without the proposed technique. Without such comparisons, it is difficult to confirm whether MTAP itself provides a performance improvement.

---

> ### Author Rebuttal · Authors · 2026-03-31
>
> We thank the reviewer for the thoughtful feedback. Below we address each concern with new experiments and clarifications.
>
> ## Response to W1: Task Complexity
>
> We agree that stronger task settings are important. Following the reviewer's suggestion, we ran new experiments on **RoboCasa-Kitchen**, which is substantially more challenging than CALVIN or SimplerEnv. We set the action chunk size to 32 to handle the longer horizons:
>
> | Method | # Demos/Task | Avg SR (%) |
> |---|---|---|
> | GR00T-N1 + DreamGen | 300 (+10k synthetic) | 57.6 |
> | GR00T-N1 + DUST | 300 | 58.5 |
> | π0 | 300 | 62.5 |
> | GR00T-N1.5 | 300 | 64.1 |
> | **RoboOmni(FAST)** | **300** | **68.2** |
>
> RoboOmni(FAST) achieves the best result, outperforming π0 by +5.7% and GR00T-N1.5 by +4.1% without synthetic augmentation. This shows that the full method remains effective at chunk size 32.
>
> ## Response to W2: Novelty of Individual Components
>
> Our contribution is not that every ingredient is new in isolation, but that **MTAP and the resulting unified discrete training recipe are novel and effective together**:
>
> 1. **MTAP is not standard multi-token prediction.** In language modeling, multi-token prediction predicts future tokens in a 1D text sequence. Here it predicts a structured future **action chunk**. For Bin tokenization, MTAP acts as temporal compression; for FAST, it serves as an auxiliary objective for frequency-domain token modeling.
>
> 2. **Octo's "interleaved" training is different from ours.** Octo interleaves data at the data-loading level and uses a diffusion action head. It does not place vision, language, and action tokens in a shared autoregressive sequence with a next-token loss. RoboOmni does.
>
> 3. **The key novelty is the integration.** MTAP's parallel action prediction is what enables efficient interleaved window packing, cross-trajectory sequence packing, VLM co-training, and CFG — all within the native VLM pipeline. Table 3 and Table 5 in our paper demonstrate that each component contributes synergistically. This integrated design is what enables the unified discrete framework to surpass continuous/flow-matching baselines, which is the core contribution.
>
> ## Response to W3: Overclaim Regarding Discrete Token Integration
>
> We agree that our original wording was too broad. We do **not** claim to be the first to model actions as discrete tokens. Our claim is narrower:
>
> > **Previous unified discrete approaches underperformed compared to continuous/flow-matching methods. RoboOmni, through MTAP and its integrated training strategies, demonstrates that a unified discrete framework can surpass the evaluated flow-matching and diffusion-based approaches.**
>
> Evidence on CALVIN ABC→D (out-of-distribution):
>
> | Method | Type | Top 1 | Top 5 | Avg Len |
> |---|---|---|---|---|
> | π0 | Flow matching | 93.8% | 59.9% | 3.84 |
> | π0.5 | Flow matching | 94.8% | 64.3% | 3.97 |
> | GR00T N1 | Diffusion | 94.2% | 66.8% | 4.01 |
> | π0-FAST | Discrete (AR) | 98.9% | 69.8% | 4.24 |
> | OpenVLA-OFT | Discrete (OFT) | 95.7% | 70.7% | 4.12 |
> | **RoboOmni(FAST)** | **Discrete + MTAP** | **99.2%** | **73.5%** | **4.35** |
>
> RoboOmni(FAST) improves Top 5 by +13.6% over π0, +9.2% over π0.5, +6.7% over GR00T N1, and +2.8% over OpenVLA-OFT. It also outperforms RoboVLMs on ABCD→D (88.1% vs. 82.6%; Table 1). We will revise the paper accordingly.
>
> ## Response to W4: Lack of Baselines
>
> We address both baseline concerns below.
>
> 1. **Expanded recent baselines.** The requested CALVIN ABC→D baselines have been added, as shown above in W3.
>
> 2. **Naive Bin/FAST without MTAP.** This ablation was already included in Table 3:
>
> | MTAP | Tokenizer | Top 1 | Top 5 | Avg Len | Speed (ms/action) |
> |---|---|---|---|---|---|
> | ✗ | FAST | 99.0% | 80.1% | 4.52 | 24.2 |
> | ✗ | BIN | 99.0% | 67.9% | 4.24 | 107 |
> | ✓ | FAST | 99.7% | 88.1% | 4.73 | 17.5 |
> | ✓ | BIN | 99.7% | 83.4% | 4.64 | 12.1 |
>
> MTAP improves Top 5 by +8.0% for FAST and +15.5% for Bin, while also improving inference speed. This confirms that the gain is not from tokenization alone.
>
> ## Response to Question: Empirical Benefits of Multi-Modal Co-Training
>
> Yes. We have direct ablation evidence on CALVIN, and the real-world results are consistent with the same trend.
>
> On CALVIN (Table 5), removing VLM co-training reduces RoboOmni(Bin) Top 5 from 83.4% to 80.6%:
>
> | Setting | Top 5 | Avg Len |
> |---|---|---|
> | Full RoboOmni(Bin) | 83.4% | 4.64 |
> | Without VLM Dataset co-training | 80.6% | 4.53 |
>
> In real-world evaluation (Figure 3), RoboOmni reaches 91% on Unseen Objects, versus 61% for π0-FAST:
>
> | Method | Unseen Distractors | Unseen Instructions | Unseen Objects |
> |---|---|---|---|
> | π0-FAST | 68% | 63% | 61% |
> | **RoboOmni** |  **89%** | **89%** | **91%** |
>
> Although this is not an isolated ablation, it is consistent with the hypothesis that co-training improves transferable visual-semantic understanding rather than only in-distribution action fitting.

---

> > ### Author Rebuttal · Reviewer_ivp2 · 2026-04-03
> >
> > While most of my concerns have been addressed, the baselines remain insufficient.
> >
> > The authors should include comparisons with multi-token prediction baselines under the same RoboOmni setting. Simply comparing against methods such as OpenVLA-OFT is not sufficient, as differences in backbone architecture, training pipeline and pretraining data distribution make it difficult to isolate whether the observed improvements stem from the multi-token prediction itself.
> >
> > As such, comparisons with alternative approaches that also generate multiple tokens under a controlled setting are still lacking.

---

> > > ### Author Response · Authors · 2026-04-04
> > >
> > > We thank the reviewer for the helpful clarification. We agree that comparisons to external methods alone (e.g., OpenVLA-OFT) do not cleanly isolate the effect of the action prediction mechanism, since they also differ in backbone, training pipeline, and data mixture. To address this concern, we implemented two representative baselines within the RoboOmni framework that also predict multiple future action tokens, and evaluated them under matched settings.
> > >
> > > The controlled baselines are:
> > >
> > > 1. **OFT-style parallel decoding**: zero-initialized action tokens attend bidirectionally, and the full action chunk is predicted in a single forward pass.
> > > 2. **π0.5-style flow matching**: an action generator predicts the full action chunk through iterative flow-matching denoising.
> > >
> > > All three methods use:
> > >
> > > - **The same backbone**: Qwen2.5-VL-7B
> > > - **The same data**: identical CALVIN ABCD→D training split
> > > - **The same VLM co-training ratio**
> > > - **The same effective CALVIN frame budget per training step**
> > > - **The same action chunk size**: H = 10
> > >
> > > Results on CALVIN ABCD→D:
> > >
> > > | Action Prediction Method | Top 1 | Top 2 | Top 3 | Top 4 | Top 5 | Avg Len |
> > > |---|---|---|---|---|---|---|
> > > | OFT-style (zero emb + full attn mask) | 98.4% | 95.2% | 91.1% | 87.5% | 81.9% | 4.54 |
> > > | π0.5-style (flow-matching action expert) | 98.5% | 95.5% | 92.3% | 88.3% | 83.6% | 4.58 |
> > > | **MTAP (ours)** | **99.7%** | **98.2%** | **95.1%** | **91.8%** | **88.1%** | **4.73** |
> > >
> > > Under these matched settings, MTAP outperforms both alternatives:
> > > - vs. OFT-style: **+6.2 points** in Top 5 (+0.19 Avg Len)
> > > - vs. π0.5-style flow matching: **+4.5 points** in Top 5 (+0.15 Avg Len)
> > >
> > > Together with the no-MTAP Bin/FAST ablations already reported in Table 3, these within-framework comparisons provide much cleaner evidence that the gain comes from MTAP itself rather than from differences in backbone, data, or training recipe.

---

### Official Review · Reviewer_TCrB · 2026-03-15

**Soundness:** 2
**Presentation:** 2
**Significance:** 3
**Originality:** 2
**Overall Recommendation:** 4
**Confidence:** 4

**Summary:**

This paper presents RoboOmni, a unified multi-modal next-token prediction framework for Vision-Language-Action (VLA) models. The central claim is that robotic actions can be effectively modeled as discrete tokens within a standard VLM pipeline, rather than requiring decoupled continuous policy heads (e.g., diffusion or flow matching). The key technical contribution is Multi-Token Action Prediction (MTAP), which enables parallel decoding of action chunks through replicated final-layer heads, resolving compounding error and inference latency issues in autoregressive action generation. The model is built on a Qwen2.5-VL backbone with multi-modal co-training tasks (VQA, visual grounding, trace prediction). Evaluations on CALVIN, SimplerEnv, and a real-robot platform show strong results.

**Compliance With Llm Reviewing Policy:**

Affirmed.

**Final Justification:**

Updated the score as the rebuttal has addressed the reviewer's concerns.

**Key Questions For Authors:**

1. How much of the performance gap over π0-FAST is attributable to the Qwen2.5-VL-7B backbone being larger and more capable than PaliGemma-3B, versus the MTAP mechanism itself? Have you run π0-FAST on the same Qwen2.5-VL backbone to control for this?
2. Does the choice of VLM backbone matter significantly—for instance, would upgrading to Qwen3-VL change the results or conclusions?
3. How does RoboOmni compare against flow-matching approaches like π0 (not π0-FAST), π0.5, Grootn1.5, etc on the same benchmarks?

**Limitations:**

yes

**Strengths And Weaknesses:**

Strengths:
1. The ablation studies are thorough, isolating the contributions of MTAP, tokenizer choice, bin size, window size, model scale, and each training strategy.
2. The inference speed analysis is compelling—MTAP with the Bin tokenizer reduces latency from 107 ms/action to 12.1 ms/action, making the unified discrete approach competitive.
3. The real-robot evaluation is reasonable, covering four generalization settings (simple, unseen distractors, unseen instructions, unseen objects) across 30+ tasks, and the margins over baselines are substantial.

Weakness:
1. The statement in Section 3.1 that "FAST tokenization inherently performs lossy compression via the Fourier transform" is factually incorrect—both the Fourier transform and the DCT used by FAST are lossless; the lossy step in FAST comes from the quantization/binning applied after the DCT, not from the transform itself, which undermines the paper's technical framing of why MTAP plays different roles for the two tokenizers.
2. The architectural differences from π0-FAST appear to reduce primarily to parallel action token decoding via MTAP and a different backbone (Qwen2.5-VL vs. PaliGemma), yet the paper claims the 8% improvement on CALVIN "proves" gains stem from architectural design; this conflates backbone capacity differences with architectural novelty, and the claim is not adequately disentangled.
3. The baseline comparisons omit several important recent models such as π0.5, Groot N1.5/N1.6, and other flow-matching approaches, which weakens the paper's argument that discrete sequence modeling should be preferred over continuous/flow-matching methods—without these comparisons, the claimed superiority is not fully justified.
> Quote from intro paragraphs: "Most VLAs apply VLMs as their feature extractors and feed representations into a decoupled continuous policy head, e.g., diffusion or flow policies (Team et al., 2024; Liu et al., 2024b), for action prediction. Although being effective for modeling continuous spaces, the decoupled approach separates action generation from core VLM reasoning and deviates from the pretrained internet-scale data."

---

> ### Author Rebuttal · Authors · 2026-03-31
>
> We thank reviewer for the detailed feedback. Below we address each concern with new controlled experiments and expanded baselines.
>
> ## Response to W1: FAST Technical Description
>
> The reviewer is correct. In FAST, the DCT itself is lossless and invertible; the lossy step comes from the subsequent quantization/binning. We will revise Section 3.1 to:
>
> > *"FAST tokenization performs lossy compression via **quantization applied after the DCT**, yielding a variable-length frequency-domain token sequence."*
>
> This correction does not affect our core claim. The different roles of MTAP come from sequence structure, not transform lossiness: FAST yields a variable-length **frequency-domain** sequence without explicit stepwise temporal alignment, whereas Bin tokenization preserves a fixed-length **temporal** sequence. Accordingly, MTAP acts as a temporal compressor for Bin and as an auxiliary objective for FAST.
>
> ## Response to W2: Disentangling Backbone vs. MTAP Contribution
>
> We reimplemented π0-FAST on the same **Qwen2.5-VL-7B backbone** as RoboOmni, enabling a controlled comparison on CALVIN ABCD→D:
>
> | Configuration | Backbone | MTAP | Training Strategies | Top 1 | Top 5 | Avg Len |
> |---|---|---|---|---|---|---|
> | π0-FAST (original) | PaliGemma-3B | ✗ | ✗ | 97.4% | 80.3% | 4.45 |
> | π0-FAST (re-impl.) | Qwen2.5-VL-7B | ✗ | ✗ | 99.4% | 76.9% | 4.44 |
> | RoboOmni w/o MTAP | Qwen2.5-VL-7B | ✗ | ✓ | 99.0% | 80.1% | 4.52 |
> | **RoboOmni (FAST)** | **Qwen2.5-VL-7B** | **✓** | **✓** | **99.7%** | **88.1%** | **4.73** |
>
> **Attribution breakdown** (See Table 5):
>
> | Factor | Δ Top 5 |
> |---|---|
> | Training strategies | +3.2% |
> | **MTAP** | **+8.0%** |
> | **Total improvement** | **+11.2%** |
>
> 1. **Backbone choice does not explain the main gain.** Replacing PaliGemma-3B with Qwen2.5-VL-7B in π0-FAST improves Top 1, but leaves Avg Len essentially unchanged (4.45 → 4.44) and lowers Top 5 (80.3% → 76.9%). Since our main claim concerns long-horizon performance, the backbone swap alone does not explain the observed gain. A likely reason is that π0-FAST uses **single-frame input** and therefore cannot benefit from Qwen2.5-VL's video pretraining and temporal modeling.
>
> 2. **MTAP is the dominant contributor.** With the backbone fixed at Qwen2.5-VL-7B, MTAP adds +8.0% Top 5 (80.1% → 88.1%), 2.5× the combined gain from all other training strategies (+3.2%).
>
> We also tested RoboOmni(FAST) with a **different VLM family**:
>
> | Backbone | Family | Params | Top 1 | Top 5 | Avg Len |
> |---|---|---|---|---|---|
> | Qwen2.5-VL-7B | Qwen2.5 | 7B | 99.7% | 88.1% | 4.73 |
> | Qwen3-VL-4B | Qwen3 | 4B | 100.0% | 87.0% | 4.72 |
>
> Qwen3-VL-4B reaches 87.0% Top 5, only 1.1% below Qwen2.5-VL-7B despite roughly half the parameters. This suggests that the gain mainly comes from MTAP and the training recipe, not backbone scale.
>
> ## Response to W3 & Q3: Missing Baselines and Flow-Matching Comparison
>
> We reproduced the requested recent baselines on CALVIN ABC→D (out-of-distribution), using official code when available and otherwise our reimplementations under the same training and evaluation protocol:
>
> | Method | Type | Top 1 | Top 5 | Avg Len |
> |---|---|---|---|---|
> | π0 | Flow matching | 93.8% | 59.9% | 3.84 |
> | π0.5 | Flow matching | 94.8% | 64.3% | 3.97 |
> | GR00T N1 | Diffusion | 94.2% | 66.8% | 4.01 |
> | π0-FAST | Discrete (AR) | 98.9% | 69.8% | 4.24 |
> | RoboVLMs | Continuous | 98.0% | 70.4% | 4.25 |
> | OpenVLA-OFT | Discrete (OFT) | 95.7% | 70.7% | 4.12 |
> | **RoboOmni(Bin)** | **Discrete + MTAP** | **98.8%** | **72.1%** | **4.30** |
> | **RoboOmni(FAST)** | **Discrete + MTAP** | **99.2%** | **73.5%** | **4.35** |
>
> RoboOmni(FAST) outperforms **π0** by +13.6%, **π0.5** by +9.2%, **GR00T N1** by +6.7%, and **OpenVLA-OFT** by +2.8% in Top 5. These results support our claim that discrete sequence modeling with MTAP outperforms the evaluated flow-matching and diffusion baselines on CALVIN ABC→D.
>
> Regarding scope, π0.5 reports only real-world mobile manipulation results, with no public simulation benchmark numbers, and newer GR00T variants target humanoid platforms with different embodiments and task distributions. Using CALVIN ABC→D as a common benchmark therefore provides the fairest direct comparison.
>
> ## Response to Q1: Performance Gap Attribution
>
> Using the re-implemented Qwen2.5-VL-7B π0-FAST baseline from W2, the backbone is held constant. The remaining +11.2% Top 5 gain to RoboOmni(FAST) decomposes into +3.2% from training strategies and +8.0% from MTAP.
>
> ## Response to Q2: VLM Backbone Sensitivity
>
> The Qwen3-VL-4B result in W2 reaches 87.0% Top 5, only 1.1% below Qwen2.5-VL-7B. Our conclusion is therefore not tied to a specific VLM family.

---

> > ### Author Rebuttal · Reviewer_TCrB · 2026-04-03
> >
> > The rebuttal has addressed the reviewer's concerns.

---

### Decision · Program_Chairs · 2026-04-30

**Decision:**

Accept (regular)

**Comment:**

RoboOmni: Vision-Language-Action model that treats robotic actions as discrete tokens. Introduces Multi-Token Action Prediction (MTAP), which integrates action chunking directly into the discrete tokenizer to enable parallel decoding, resolving temporal modeling bottlenecks and compounding errors.

Rating: Reviewer TCrB (4), Reviewer ivp2 (4), Reviewer yUN2 (4), Reviewer WvzF (3). The reviewer likes the clear and unified formation which helps simplify the integration of VLM’s capabilities and robot control. They find MTAP sound and empirical performance strong (CALVIN, SimplerEnv, real-world setups) both in terms of reducing compounding effects and inference latency.

Major Concerns

1. [Reviewer TCrB, Reviewer ivp2, Reviewer WvzF, resolved] Lack of strong baselines. Reviewer TCrB and Reviewer WvzF mention strong SOTA models including π0.5, GROOT variants, OpenVLA-OFT, and SpatialVLA. Reviewer ivp2 wants to see baselines trained with Bin or FAST tokenizers without MTAP. The authors reproduced and provided results for π0, π0.5, GROOT N1, and OpenVLA-OFT on the CALVIN ABC->D benchmark, which show RoboOmni outperforming them. The author further validated the benefit of MTAP using controlled internal baselines (OFT-style parallel decoding and π0.5-style flow-matching) under matched settings to. Reviewer WvzF still had some concern over training fairness for SOTA models, but the authors clarified their reproduced numbers align with other third-party papers and that that no official CALVIN results exist for those models.

2. [Reviewer TCrB, Reviewer WvzF, resolved] Backbone. The reviewers question if the performance gains come from a stronger backbone (Qwen2.5-VL-7B). The authors re-implemented π0-FAST on the same Qwen2.5-VL-7B backbone and evaluated RoboOmni on a smaller Qwen3-VL-4B backbone, and still reported positive results.

3. [Reviewer ivp2, Reviewer yUN2, Reviewer WvzF, resolved] Discretization. The reviewers question the limit of discretization for complex or fine tasks. The authors provided a result on RoboCasa-Kitchen during rebuttal and revised their discussion.

4. [Reviewer WvzF, resolved] Error analysis. The authors identified two primary failure modes and promised to add qualitative examples.

The authors resolved other concerns by revising their claims, providing a direct ablation justification, or pointing to the results in the paper/appendix. Overall, concerns are sufficiently addressed.